# Graphon Neural Networks and the Transferability of Graph Neural Networks

**Luana Ruiz**
Dept. of Electrical and Systems Eng.
University of Pennsylvania
Philadelphia, PA 19143
`rubruiz@seas.upenn.edu`

**Luiz F. O. Chamon**
Dept. of Electrical and Systems Eng.
University of Pennsylvania
Philadelphia, PA 19143
`luizf@seas.upenn.edu`

**Alejandro Ribeiro**
Dept. of Electrical and Systems Eng.
University of Pennsylvania
Philadelphia, PA 19143
`aribeiro@seas.upenn.edu`

## Abstract

Graph neural networks (GNNs) rely on graph convolutions to extract local features from network data. These graph convolutions combine information from adjacent nodes using coefficients that are shared across all nodes. Since these coefficients are shared and do not depend on the graph, one can envision using the same coefficients to define a GNN on another graph. This motivates analyzing the transferability of GNNs across graphs. In this paper we introduce graphon NNs as limit objects of GNNs and prove a bound on the difference between the output of a GNN and its limit graphon-NN. This bound vanishes with growing number of nodes if the graph convolutional filters are bandlimited in the graph spectral domain. This result establishes a tradeoff between discriminability and transferability of GNNs.

## 1 Introduction

Graph neural networks (GNNs) are the counterpart of convolutional neural networks (CNNs) to learning problems involving network data. Like CNNs, GNNs have gained popularity due to their superior performance in a number of learning tasks (Bronstein et al., 2017; Defferrard et al., 2016; Gama et al., 2018; Kipf and Welling, 2017). Aside from the ample amount of empirical evidence, GNNs are proven to work well because of properties such as invariance and stability (Gama et al., 2019a; Ruiz et al., 2019), which are also shared with CNNs (Bruna and Mallat, 2013).

A defining characteristic of GNNs is that their number of parameters does not depend on the size (i.e., the number of nodes) of the underlying graph. This is because graph convolutions are parametrized by graph shifts in the same way that time and spatial convolutions are parametrized by delays and translations. From a complexity standpoint, the independence between the GNN parametrization and the graph is beneficial because there are less parameters to learn. Perhaps more importantly, the fact that its parameters are not tied to the underlying graph suggests that a GNN can be transferred from graph to graph. It is then natural to ask to what extent the performance of a GNN is preserved when its graph changes. The ability to transfer a machine learning model with performance guarantees is usually referred to as transfer learning or *transferability*.

In GNNs, there are two typical scenarios where transferability is desirable. The first involves applications in which we would like to reproduce a previously trained model on a graph of different

size, but similar to the original graph in a sense that we formalize in this paper, with performance guarantees. This is useful when we cannot afford to retrain the model. The second concerns problems where the network size changes over time. In this scenario, we would like the GNN model to be robust to nodes being added or removed from the network, i.e., for it to be transferable in a scalable way. An example are recommender systems based on a growing user network.

Both of these scenarios involve solving the same task on networks that, although different, can be seen as being of the same "type". This motivates studying the transferability of GNNs within families of graphs that share certain structural characteristics. We propose to do so by focusing on collections of graphs associated with the same *graphon*. A graphon is a bounded symmetric kernel $\mathbf{W} : [0,1]^2 \to [0,1]$ that can be interpreted a a graph with an uncountable number of nodes. Graphons are suitable representations of graph families because they are the limit objects of sequences of graphs where the density of certain structural "motifs" is preserved. They can also be used as generating models for undirected graphs where, if we associate nodes $i$ and $j$ with points $u_i$ and $u_j$ in the unit interval, $\mathbf{W}(u_i, u_j)$ is the weight of the edge $(i, j)$. The main result of this paper (Theorem 2) shows that GNNs are transferable between deterministic graphs obtained from a graphon in this way.

**Theorem** (GNN transferability, informal) Let $\mathbf{\Phi}(\mathbf{G})$ be a GNN with fixed parameters. Let $\mathbf{G}_{n_1}$ and $\mathbf{G}_{n_2}$ be deterministic graphs with $n_1$ and $n_2$ nodes obtained from a graphon $\mathbf{W}$. Under mild conditions, $\|\mathbf{\Phi}(\mathbf{G}_{n_1}) - \mathbf{\Phi}(\mathbf{G}_{n_2})\| = \mathcal{O}(n_1^{-0.5} + n_2^{-0.5})$.

An important consequence of this result is the existence of a trade-off between transferability and discriminability for GNNs on these deterministic graphs, which is related to a restriction on the passing band of the graph convolutional filters of the GNN. Its proof is based on the definition of the *graphon neural network* (Section 4), a theoretical limit object of independent interest that can be used to generate GNNs on deterministic graphs from a common family. The interpretation of graphon neural networks as generating models for GNNs is important because it identifies the graph as a flexible parameter of the learning architecture and allows adapting the GNN not only by changing its weights, but also by changing the underlying graph. While this result only applies to deterministic graphs instantiated from graphons (i.e., it does not encompass stochastic graphs sampled from the graphon, or sparser graphs, which are better modeled by *graphings* (Lovász, 2012, Chapter 18)), it provides useful insights on GNNs and on their transferability properties.

The rest of this paper is organized as follows. Section 2 goes over related work. Section 3 introduces preliminary definitions and discusses GNNs and graphon information processing. The aforementioned contributions are presented in Sections 4 and 5. In Section 6, transferability of GNNs is illustrated in two numerical experiments. Concluding remarks are presented in Section 7, and proofs and additional numerical experiments are deferred to the supplementary material.

## 2   Related Work

Graphons and convergent graph sequences have been broadly studied in mathematics (Borgs et al., 2008, 2012; Lovász, 2012; Lovász and Szegedy, 2006) and have found applications in statistics (Gao et al., 2015; Wolfe and Olhede, 2013; Xu, 2018), game theory (Parise and Ozdaglar, 2019), network science (Avella-Medina et al., 2018; Vizuete et al., 2020) and controls (Gao and Caines, 2020). Recent works also use graphons to study network information processing in the limit (Morency and Leus, 2017; Ruiz et al., 2020; Ruiz et al., 2020). In particular, Ruiz et al. (2020) study the convergence of graph signals and graph filters by introducing the theory of signal processing on graphons. The use of limit and continuous objects, e.g. neural tangent models (Jacot et al., 2018), is also common in the analysis of the behavior of neural networks.

A concept related to transferability is the notion of stability of GNNs to graph perturbations. This is studied in (Gama et al., 2019a) building on stability analyses of graph scattering transforms (Gama et al., 2019b). the stability of GNNs on standard These results do not consider graphs of varying size. In contrast, Keriven et al. (2020) study stability of GNNs on random graphs. Transferability as the number of nodes in a graph grows is analyzed in (Levie et al., 2019a), following up on the work of Levie et al. (2019b) which studies the transferability of spectral graph filters. This work looks at graphs as discretizations of generic topological spaces, which yields a different asymptotic regime relative to the graphon limits we consider in this paper.

# 3 Preliminary Definitions

We go over the basic architecture of a graph neural network and formally introduce graphons and graphon data. These concepts will be important in the definition of graphon neural networks in Section 4 and in the derivation of a transferability bound for GNNs in Section 5.

## 3.1 Graph neural networks

GNNs are deep convolutional architectures with two main components per layer: a bank of graph convolutional filters or *graph convolutions*, and a nonlinear activation function. The graph convolution couples the data with the underlying network, lending GNNs the ability to learn accurate representations of network data.

Networks are represented as graphs $\mathbf{G} = (\mathcal{V}, \mathcal{E}, w)$, where $\mathcal{V}$, $|\mathcal{V}| = n$, is the set of nodes, $\mathcal{E} \subseteq \mathcal{V} \times \mathcal{V}$ is the set of edges and $w : \mathcal{E} \to \mathbb{R}$ is a function assigning weights to the edges of $\mathbf{G}$. We restrict attention to undirected graphs, so that $w(i, j) = w(j, i)$. Network data are modeled as *graph signals* $\mathbf{x} \in \mathbb{R}^n$, where each element $[\mathbf{x}]_i = x_i$ corresponds to the value of the data at node $i$ (Ortega et al., 2018; Shuman et al., 2013). In this setting, it is natural to model data exchanges as operations parametrized by the graph. This is done by considering the graph shift operator (GSO) $\mathbf{S} \in \mathbb{R}^{n \times n}$, a matrix that encodes the sparsity pattern of $\mathbf{G}$ by satisfying $[\mathbf{S}]_{ij} = s_{ij} \neq 0$ only if $i = j$ or $(i, j) \in \mathcal{E}$. In this paper, we use the adjacency matrix $[\mathbf{A}]_{ij} = w(i, j)$ as the GSO, but other examples include the degree matrix $\mathbf{D} = \text{diag}(\mathbf{A1})$ and the graph Laplacian $\mathbf{L} = \mathbf{D} - \mathbf{A}$.

The GSO effects a *shift*, or diffusion, of data on the network. Note that, at each node $i$, the operation $\mathbf{Sx}$ is given by $\sum_{j|(i,j)\in\mathcal{E}} s_{ij}x_j$, i.e., nodes $j$ shift their data values to neighbors $i$ according to their proximity measured by $s_{ij}$. This notion of shift allows defining the convolution operation on graphs. In time or space, the filter convolution is defined as a weighted sum of data shifted through delays or translations. Analogously, we define the graph convolution as a weighted sum of data shifted to neighbors at most $K - 1$ hops away. Explicitly,

$$\mathbf{h} *_{\mathbf{S}} \mathbf{x} = \sum_{k=0}^{K-1} h_k \mathbf{S}^k \mathbf{x} = \mathbf{H}(\mathbf{S})\mathbf{x} \tag{1}$$

where $\mathbf{h} = [h_0, \dots h_{K-1}]$ are the filter coefficients and $*_{\mathbf{S}}$ denotes the convolution operation with GSO $\mathbf{S}$. Because the graph is undirected, $\mathbf{S}$ is symmetric and diagonalizable as $\mathbf{S} = \mathbf{V}\mathbf{\Lambda}\mathbf{V}^{\mathsf{H}}$, where $\mathbf{\Lambda}$ is a diagonal matrix containing the graph eigenvalues and $\mathbf{V}$ forms an orthonormal eigenvector basis that we call the graph spectral basis. Replacing $\mathbf{S}$ by its diagonalization in (1) and calculating the change of basis $\mathbf{V}^{\mathsf{H}}\mathbf{H}(\mathbf{S})\mathbf{x}$, we get

$$\mathbf{V}^{\mathsf{H}}\mathbf{H}(\mathbf{S})\mathbf{x} = \sum_{k=0}^{K-1} h_k \mathbf{\Lambda}^k \mathbf{V}^{\mathsf{H}}\mathbf{x} = h(\mathbf{\Lambda})\mathbf{V}^{\mathsf{H}}\mathbf{x} \tag{2}$$

from which we conclude that the graph convolution $\mathbf{H}(\mathbf{S})$ has a spectral representation $h(\lambda) = \sum_{k=0}^{K-1} h_k \lambda^k$ which only depends on the coefficients $\mathbf{h}$ and on the eigenvalues of $\mathbf{G}$.

Denoting the nonlinear activation function $\rho$, the $\ell$th layer of a GNN is written as

$$\mathbf{x}_\ell^f = \rho \left( \sum_{g=1}^{F_{\ell-1}} \mathbf{h}_\ell^{fg} *_{\mathbf{S}} \mathbf{x}_{\ell-1}^g \right) \tag{3}$$

for each feature $\mathbf{x}_\ell^f$, $1 \leq f \leq F_\ell$. The quantities $F_{\ell-1}$ and $F_\ell$ are the numbers of features at the output of layers $\ell - 1$ and $\ell$ respectively for $1 \leq \ell \leq L$. The GNN output is $\mathbf{y}^f = \mathbf{x}_L^f$, while the input features at the first layer, which we denote $\mathbf{x}_0^g$, are the input data $\mathbf{x}^g$ with $1 \leq g \leq F_0$. For a more succinct representation, this GNN can also be expressed as a map $\mathbf{y} = \mathbf{\Phi}(\mathcal{H}; \mathbf{S}; \mathbf{x})$, where the set $\mathcal{H}$ groups all learnable parameters $\mathbf{h}_\ell^{fg}$ as $\mathcal{H} = \{\mathbf{h}_\ell^{fg}\}_{\ell,f,g}$.

In (3), note that the GNN parameters $\mathbf{h}_\ell^{fg}$ do not depend on $n$, the number of nodes of $\mathbf{G}$. This means that, once the model is trained and these parameters are learned, the GNN can be used to perform inference on any other graph by replacing $\mathbf{S}$ in (3). In this case, the goal of transfer learning is for the

model to maintain a similar enough performance in the same task over different graphs. A question that arises is then: for which graphs are GNNs transferable? To answer this question, we focus on graphs belonging to "graph families" identified by graphons.

## 3.2 Graphons and graphon data

A graphon is a bounded, symmetric, measurable function $\mathbf{W} : [0,1]^2 \to [0,1]$ that can be thought of as an undirected graph with an uncountable number of nodes. This can be seen by relating nodes $i$ and $j$ with points $u_i, u_j \in [0,1]$, and edges $(i,j)$ with weights $\mathbf{W}(u_i, u_j)$. This construction suggests a limit object interpretation and, in fact, it is possible to define sequences of graphs $\{\mathbf{G}_n\}_{n=1}^{\infty}$ that converge to $\mathbf{W}$.

### 3.2.1 Graphons as limit objects

To characterize the convergence of a graph sequence $\{\mathbf{G}_n\}$, we consider arbitrary unweighted and undirected graphs $\mathbf{F} = (\mathcal{V}', \mathcal{E}')$ that we call "graph motifs". Homomorphisms of $\mathbf{F}$ into $\mathbf{G} = (\mathcal{V}, \mathcal{E}, w)$ are adjacency preserving maps in which $(i,j) \in \mathcal{E}'$ implies $(i,j) \in \mathcal{E}$. There are $|\mathcal{V}|^{|\mathcal{V}'|} = n^{n'}$ maps from $\mathcal{V}'$ to $\mathcal{V}$, but only some of them are homomorphisms. Hence, we can define a density of homomorphisms $t(\mathbf{F}, \mathbf{G})$, which represents the relative frequency with which the motif $\mathbf{F}$ appears in $\mathbf{G}$.

Homomorphisms of graphs into graphons are defined analogously. Denoting $t(\mathbf{F}, \mathbf{W})$ the density of homomorphisms of the graph $\mathbf{F}$ into the graphon $\mathbf{W}$, we then say that a sequence $\{\mathbf{G}_n\}$ converges to the graphon $\mathbf{W}$ if, for all finite, unweighted and undirected graphs $\mathbf{F}$,

$$\lim_{n \to \infty} t(\mathbf{F}, \mathbf{G}_n) = t(\mathbf{F}, \mathbf{W}). \tag{4}$$

It can be shown that every graphon is the limit object of a convergent graph sequence, and every convergent graph sequence converges to a graphon (Lovász, 2012, Chapter 11). Thus, a graphon identifies an entire collection of graphs. Regardless of their size, these graphs can be considered similar in the sense that they belong to the same "graphon family".

A simple example of convergent graph sequence is obtained by evaluating the graphon. In particular, in this paper we are interested in *deterministic graphs* $\mathbf{G}_n$ constructed by assigning regularly spaced points $u_i = (i-1)/n$ to nodes $1 \le i \le n$ and weights $\mathbf{W}(u_i, u_j)$ to edges $(i,j)$, i.e.

$$[\mathbf{S}_n]_{ij} = s_{ij} = \mathbf{W}(u_i, u_j) \tag{5}$$

where $\mathbf{S}_n$ is the adjacency matrix of $\mathbf{G}_n$. An example of a stochastic block model graphon and of an 8-node deterministic graph drawn from it are shown at the top of Figure 1, from left to right. A sequence $\{\mathbf{G}_n\}$ generated in this fashion satisfies the condition in (4), therefore $\{\mathbf{G}_n\}$ converges to $\mathbf{W}$ (Lovász, 2012, Chapter 11).

### 3.2.2 Graphon information processing

Data on graphons can be seen as an abstraction of network data on graphs with an uncountable number of nodes. Graphon data is defined as graphon signals $X \in L_2([0,1])$ mapping points of the unit interval to the real numbers (Ruiz et al., 2020). The coupling between this data and the graphon is given by the integral operator $T_{\mathbf{W}} : L_2([0,1]) \to L_2([0,1])$, which is defined as

$$(T_{\mathbf{W}}X)(v) := \int_0^1 \mathbf{W}(u,v)X(u)du. \tag{6}$$

Since $\mathbf{W}$ is bounded and symmetric, $T_{\mathbf{W}}$ is a self-adjoint Hilbert-Schmidt operator. This allows expressing the graphon in the operator's spectral basis as $\mathbf{W}(u,v) = \sum_{i \in \mathbb{Z} \setminus \{0\}} \lambda_i \varphi_i(u) \varphi_i(v)$ and rewriting $T_{\mathbf{W}}$ as

$$(T_{\mathbf{W}}X)(v) = \sum_{i \in \mathbb{Z} \setminus \{0\}} \lambda_i \varphi_i(v) \int_0^1 \varphi_i(u)X(u)du \tag{7}$$

where the eigenvalues $\lambda_i$, $i \in \mathbb{Z} \setminus \{0\}$, are ordered according to their sign and in decreasing order of absolute value, i.e. $1 \ge \lambda_1 \ge \lambda_2 \ge \dots \ge \dots \ge \lambda_{-2} \ge \lambda_{-1} \ge -1$, and accumulate around 0 as $|i| \to \infty$ (Lax, 2002, Theorem 3, Chapter 28).

Similarly to the GSO, $T_{\mathbf{W}}$ defines a notion of shift on the graphon. We refer to it as the graphon shift operator (WSO), and use it to define the graphon convolution as a weighted sum of at most $K-1$ data shifts. Explicitly,

$$\mathbf{h} *_{\mathbf{W}} X = \sum_{k=0}^{K-1} h_k (T_{\mathbf{W}}^{(k)} X)(v) = (T_{\mathbf{H}} X)(v) \quad \text{with}$$

$$(T_{\mathbf{W}}^{(k)} X)(v) = \int_0^1 \mathbf{W}(u,v)(T_{\mathbf{W}}^{(k-1)} X)(u)du \tag{8}$$

where $T_{\mathbf{W}}^{(0)} = \mathbf{I}$ is the identity operator (Ruiz et al., 2020). The operation $*_{\mathbf{W}}$ stands for the convolution with graphon $\mathbf{W}$, and $\mathbf{h} = [h_0, \ldots, h_{K-1}]$ are the filter coefficients. Using the spectral decomposition in (7), $T_{\mathbf{H}}$ can also be written as

$$(T_{\mathbf{H}} X)(v) = \sum_{i \in \mathbb{Z} \setminus \{0\}} \sum_{k=0}^{K-1} h_k \lambda_i^k \varphi_i(v) \int_0^1 \varphi_i(u) X(u)du = \sum_{i \in \mathbb{Z} \setminus \{0\}} h(\lambda_i) \varphi_i(v) \int_0^1 \varphi_i(u) X(u)du \tag{9}$$

where we note that, like the graph convolution, $T_{\mathbf{H}}$ has a spectral representation $h(\lambda) = \sum_{k=0}^{K-1} h_k \lambda^k$ which only depends on the graphon eigenvalues and the coefficients $h_k$.

## 4   Graphon Neural Networks

Similarly to how sequences of graphs converge to graphons, we can think of a sequence of GNNs converging to a graphon neural network (WNN). This limit architecture is defined by a composition of layers consisting of graphon convolutions and nonlinear activation functions, tailored to process data supported on graphons. Denoting the nonlinear activation function $\rho$, the $\ell$th layer of a graphon neural network can be written as

$$X_\ell^f = \rho \left( \sum_{g=1}^{F_{\ell-1}} \mathbf{h}_\ell^{fg} *_{\mathbf{W}} X_{\ell-1}^g \right) \tag{10}$$

for $1 \leq f \leq F_\ell$, where $F_\ell$ stands for the number of features at the output of layer $\ell$, $1 \leq \ell \leq L$. The WNN output is given by $Y^f = X_L^f$, and the input features at the first layer, $X_0^g$, are the input data $X^g$ for $1 \leq g \leq F_0$. A more succinct representation of this WNN can be obtained by writing it as the map $Y = \mathbf{\Phi}(\mathcal{H}; \mathbf{W}; X)$, where $\mathcal{H} = \{\mathbf{h}_\ell^{fg}\}_{\ell,f,g}$ groups the filter coefficients at all layers. Note that the parameters in $\mathcal{H}$ are agnostic to the graphon.

### 4.1   WNNs as deterministic generating models for GNNs

Comparing the GNN and WNN maps $\mathbf{\Phi}(\mathcal{H}; \mathbf{S}; \mathbf{x})$ [cf. Section 4] and $\mathbf{\Phi}(\mathcal{H}; \mathbf{W}; X)$, we see that they can have the same set of parameters $\mathcal{H}$. On graphs belonging to a graphon family, this means that GNNs can be built as instantiations of the WNN and, therefore, WNNs can be seen as generative models for GNNs. We consider GNNs $\mathbf{\Phi}(\mathcal{H}; \mathbf{S}_n; \mathbf{x}_n)$ built from a WNN $\mathbf{\Phi}(\mathcal{H}; \mathbf{W}; X)$ by defining $u_i = (i-1)/n$ for $1 \leq i \leq n$ and setting

$$[\mathbf{S}_n]_{ij} = \mathbf{W}(u_i, u_j) \quad \text{and}$$
$$[\mathbf{x}_n]_i = X(u_i) \tag{11}$$

where $\mathbf{S}_n$ is the GSO of $\mathbf{G}_n$, the deterministic graph obtained from $\mathbf{W}$ as in Section 3.2, and $\mathbf{x}_n$ is the *deterministic graph signal* obtained by evaluating the graphon signal $X$ at points $u_i$. An example of a WNN and of a GNN instantiated in this way are shown in Figure 1. Considering GNNs as instantiations of WNNs is interesting because it allows looking at graphs not as fixed GNN hyperparameters, but as parameters that can be tuned. I.e., it allows GNNs to be adapted both by optimizing the weights in $\mathcal{H}$ and by changing the graph $\mathbf{G}_n$ to any graph that can be deterministically evaluated from the graphon as in (11). This makes the learning model scalable and adds flexibility in cases where there are uncertainties associated with the graph but the graphon is known.

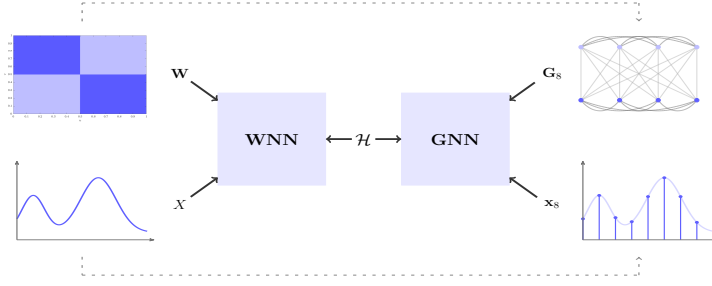

Figure 1: Example of a graphon neural network (WNN) given by $\Phi(\mathcal{H}; \mathbf{W}; X)$, and of a graph neural network (GNN) instantiated from it as $\Phi(\mathcal{H}; \mathbf{S}_8; \mathbf{x}_8)$. The graphon $\mathbf{W}$, shown on the top left corner, is a stochastic block model with intra-community probability $p_{c_i c_i} = 0.8$ and inter-community probability $p_{c_i c_j} = 0.2$, and the graphon signal $X$ is plotted on the bottom left corner. The graph $\mathbf{G}_8$ (top right corner) and the graph signal $\mathbf{x}_8$ (bottom right corner) are obtained from $\mathbf{W}$ and $X$ according to (11). Note that the parameter set $\mathcal{H}$ is shared between the WNN and the GNN.

Conversely, we can also define WNNs induced by GNNs. The WNN induced by a GNN $\Phi(\mathcal{H}; \mathbf{S}_n; \mathbf{x}_n)$ is defined as $\Phi(\mathcal{H}; \mathbf{W}_n; X_n)$, and it is obtained by constructing a partition $I_1 \cup \ldots \cup I_n$ of $[0, 1]$ with $I_i = [(i-1)/n, i/n]$ to define

$$
\begin{aligned}
\mathbf{W}_n(u, v) &= [\mathbf{S}_n]_{ij} \times \mathbb{I}(u \in I_i)\mathbb{I}(v \in I_j) \quad \text{and} \\
X_n(u) &= [\mathbf{x}_n]_i \times \mathbb{I}(u \in I_i)
\end{aligned}
\tag{12}
$$

where $\mathbf{W}_n$ is the *graphon induced by* $\mathbf{G}_n$ and $X_n$ is the *graphon signal induced by the graph signal* $\mathbf{x}_n$. This definition is useful because it allows comparing GNNs with WNNs.

## 4.2 Approximating WNNs with GNNs

Consider deterministic GNNs instantiated from a WNN as in (11). For increasing $n$, $\mathbf{G}_n$ converges to $\mathbf{W}$, so we can expect the GNNs to become increasingly similar to the WNN. In other words, the output of the GNN $\Phi(\mathcal{H}; \mathbf{S}_n; \mathbf{x}_n)$ and of the WNN $\Phi(\mathcal{H}; \mathbf{W}; X)$ should grow progressively close and $\Phi(\mathcal{H}; \mathbf{S}_n; \mathbf{x}_n)$ can be used to approximate $\Phi(\mathcal{H}; \mathbf{W}; X)$. We wish to quantify how good this approximation is for different values of $n$. Naturally, the continuous output $Y = \Phi(\mathcal{H}; \mathbf{W}; X)$ cannot be compared with the discrete output $\mathbf{y}_n = \Phi(\mathcal{H}; \mathbf{S}_n; \mathbf{x}_n)$ directly. In order to make this comparison, we consider the output of the WNN induced by $\Phi(\mathcal{H}; \mathbf{S}_n; \mathbf{x}_n)$, which is given by $Y_n = \Phi(\mathcal{H}; \mathbf{W}_n; X_n)$ [cf. (12)]. We also consider the following assumptions.

**AS1.** *The graphon $\mathbf{W}$ is $A_1$-Lipschitz, i.e. $|\mathbf{W}(u_2, v_2) - \mathbf{W}(u_1, v_1)| \le A_1(|u_2 - u_1| + |v_2 - v_1|)$.*

**AS2.** *The convolutional filters $h$ are $A_2$-Lipschitz and non-amplifying, i.e. $|h(\lambda)| < 1$.*

**AS3.** *The graphon signal $X$ is $A_3$-Lipschitz.*

**AS4.** *The activation functions are normalized Lipschitz, i.e. $|\rho(x) - \rho(y)| \le |x - y|$, and $\rho(0) = 0$.*

**Theorem 1** (WNN approximation by GNN). *Consider the $L$-layer WNN given by $Y = \Phi(\mathcal{H}; \mathbf{W}; \mathbf{X})$, where $F_0 = F_L = 1$ and $F_\ell = F$ for $1 \le \ell \le L - 1$. Let the graphon convolutions $h(\lambda)$ [cf. (9)] be such that $h(\lambda)$ is constant for $|\lambda| < c$. For the GNN instantiated from this WNN as $\mathbf{y}_n = \Phi(\mathcal{H}; \mathbf{S}_n; \mathbf{x}_n)$ [cf. (11)], under Assumptions 1 through 4 it holds*

$$
\|Y_n - Y\|_{L_2} \le LF^{L-1}\sqrt{A_1}\left(A_2 + \frac{\pi n_c}{\delta_c}\right)n^{-\frac{1}{2}}\|X\|_{L_2} + \frac{A_3}{\sqrt{3}}n^{-\frac{1}{2}}
$$

*where $Y_n = \Phi(\mathcal{H}; \mathbf{W}_n; X_n)$ is the WNN induced by $\mathbf{y}_n = \Phi(\mathcal{H}; \mathbf{S}_n; \mathbf{x}_n)$ [cf. (12)], $n_c$ is the cardinality of the set $\mathcal{C} = \{i \mid |\lambda_i^n| \ge c\}$, and $\delta_c = \min_{i \in \mathcal{C}}(|\lambda_i - \lambda_{i+sgn(i)}^n|, |\lambda_{i+sgn(i)}^n - \lambda_i^n|, |\lambda_1 - \lambda_{-1}^n|, |\lambda_1^n - \lambda_{-1}|)$, with $\lambda_i$ and $\lambda_i^n$ denoting the eigenvalues of $\mathbf{W}$ and $\mathbf{W}_n$ respectively and each index $i$ corresponding to a different eigenvalue.*

From Theorem 1, we conclude that a graphon neural network $\Phi(\mathcal{H}; \mathbf{W}; X)$ can be approximated with performance guarantees by the GNN $\Phi(\mathcal{H}; \mathbf{S}_n; \mathbf{x}_n)$ where the graph $\mathbf{G}_n$ and the signal $\mathbf{x}_n$

are deterministic and obtained from $\mathbf{W}$ and $X$ as in (11). In this case, the approximation error $\|Y_n - Y\|_{L_2}$ is controlled by the transferability constant $LF^{L-1}\sqrt{A_1}\left(A_2 + (\pi n_c)/\delta_c\right)n^{-0.5}$ and the fixed error term $A_3/\sqrt{3n}$, both of which decay with $\mathcal{O}(n^{-0.5})$.

The fixed error term only depends on the graphon signal variability $A_3$, and measures the difference between the graphon signal $X$ and the graph signal $\mathbf{x}_n$. The transferability constant depends on the graphon and on the parameters of the GNN. The dependence on the graphon is related to the Lipschitz constant $A_1$, which is smaller for graphons with less variability. The dependence on the architecture happens through the numbers of layers and features $L$ and $F$, as well as through the parameters $A_2$, $n_c$ and $\delta_c$ of the graph convolution. Although these parameters can be tuned, note that, in general, deeper and wider architectures have larger approximation error.

For better approximation, the convolutional filters $h$ should have limited variability, which is controlled by both the Lipschitz constant $A_2$ and the length of the band $[c, 1]$. The number of eigenvalues $n_c$ should satisfy $n_c \ll n$ (i.e. $n_c < \sqrt{n}$) for asymptotic convergence, which is guaranteed by the fact that the eigenvalues of $\mathbf{W}_{\mathbf{G}_n}$ converge to the eigenvalues of $\mathbf{W}$ (Lovász, 2012, Chapter 11.6) and therefore $\delta_c \to \min_{i \in \mathcal{C}} |\lambda_i - \lambda_{i+\mathrm{sgn}(i)}|$, which is the minimum eigengap of the graphon for $i \in \mathcal{C}$.

We also point out that Assumption 4 and the assumption that the convolutional filters have height $|h(\lambda)| < 1$ are not too strong; Assumption 4 holds for most conventional activation functions (e.g. ReLU, hyperbolic tangent) and $|h(\lambda)| < 1$ can be achieved by normalization.

## 5 Transferability of Graph Neural Networks

The main result of this paper is that GNNs are transferable between graphs of different sizes obtained deterministically from the same graphon [cf. (11)]. This result follows from Theorem 1 by the triangle inequality.

**Theorem 2** (GNN transferability). *Let $\mathbf{G}_{n_1}$ and $\mathbf{G}_{n_2}$, and $\mathbf{x}_{n_1}$ and $\mathbf{x}_{n_2}$, be graphs and graph signals obtained from the graphon $\mathbf{W}$ and the graphon signal $X$ as in (11), with $n_1 \neq n_2$. Consider the $L$-layer GNNs given by $\mathbf{\Phi}(\mathcal{H}; \mathbf{S}_{n_1}; \mathbf{x}_{n_1})$ and $\mathbf{\Phi}(\mathcal{H}; \mathbf{S}_{n_2}; \mathbf{x}_{n_2})$, where $F_0 = F_L = 1$ and $F_\ell = F$ for $1 \leq \ell \leq L - 1$. Let the graph convolutions $h(\lambda)$ [cf. (2)] be such that $h(\lambda)$ is constant for $|\lambda| < c$. Then, under Assumptions 1 through 4 it holds*

$$\|Y_{n_1} - Y_{n_2}\|_{L_2} \leq LF^{L-1}\sqrt{A_1}\left(A_2 + \frac{\pi n_c'}{\delta_c'}\right)\left(n_1^{-\frac{1}{2}} + n_2^{-\frac{1}{2}}\right)\|X\|_{L_2} + \frac{A_3}{\sqrt{3}}\left(n_1^{-\frac{1}{2}} + n_2^{-\frac{1}{2}}\right)$$

*where $Y_{n_j} = \mathbf{\Phi}(\mathcal{H}; \mathbf{W}_{n_j}; X_{n_j})$ is the WNN induced by $\mathbf{y}_{n_j} = \mathbf{\Phi}(\mathcal{H}; \mathbf{S}_{n_j}; \mathbf{x}_{n_j})$ [cf. (12)], $n_c' = \max_{j \in \{1,2\}} |\mathcal{C}_j|$ is the maximum cardinality of the sets $\mathcal{C}_j = \{i \mid |\lambda_i^{n_j}| \geq c\}$, and $\delta_c' = \min_{i \in \mathcal{C}_j, j \in \{1,2\}}(|\lambda_i - \lambda_{i+sgn(i)}^{n_j}|, |\lambda_{i+sgn(i)} - \lambda_i^{n_j}|, |\lambda_1 - \lambda_{-1}^{n_j}|, |\lambda_1^{n_j} - \lambda_{-1}|)$, with $\lambda_i$ and $\lambda_i^{n_j}$ denoting the eigenvalues of $\mathbf{W}$ and $\mathbf{W}_{n_j}$ respectively and each index $i$ corresponding to a different eigenvalue.*

Theorem 2 compares the vector outputs of the same GNN (with same parameter set $\mathcal{H}$) on $\mathbf{G}_{n_1}$ and $\mathbf{G}_{n_2}$ by bounding the $L_2$ norm difference between the graphon neural networks induced by $\mathbf{y}_{n_1} = \mathbf{\Phi}(\mathcal{H}; \mathbf{S}_{n_1}; \mathbf{x}_{n_1})$ and by $\mathbf{y}_{n_2} = \mathbf{\Phi}(\mathcal{H}; \mathbf{S}_{n_2}; \mathbf{x}_{n_2})$. This result is useful in two important ways. First, it means that, provided that its design parameters are chosen carefully, a GNN trained on a given deterministic graph can be transferred to other deterministic graphs in the same graphon family [cf. (11)] with performance guarantees. This is desirable in problems where the same task has to be replicated on different instances of such deterministic networks, because it may help avoid retraining the GNN on each graph. Second, it implies that GNNs are scalable, as the graph on which a GNN is trained can be smaller than the deterministic graphs on which it is deployed, and vice-versa. This is helpful in problems where the graph size may change. In this case, the advantage of transferability is mainly that training GNNs on smaller graphs is easier than training them on large graphs.

When transferring GNNs between deterministic graphs, the performance guarantee is measured by the transferability constant $LF^{L-1}\sqrt{A_1}(A_2 + \pi n_c'/\delta_c')(n_1^{-0.5} + n_2^{-0.5})$ and the fixed error term $A_3(n_1^{-0.5} + n_2^{-0.5})/\sqrt{3}$. Both of these terms decay with $\mathcal{O}(1/\sqrt{\min n_1, n_2})$, i.e., the transferability bound is dominated by the size of the smaller graph. As such, when one of the graphs is small the transferability bound may not be very good, even if the other graph is large.

The fixed error term measures the difference between the graph signals $\mathbf{x}_{n_1}$ and $\mathbf{x}_{n_2}$ through their distance to the graphon signal $X$, so its contribution is small if both signals are associated with the same data model. Importantly, the transferability constant depends implicitly on the graphon varibaility through $A_1$. It also depends on the width $F$ and depth $L$ of the GNN, and on the convolutional filter parameters $A_2$, $n_c'$ and $\delta_c'$. These are design parameters which can be tuned. In particular, if we make $n_c' < \sqrt{n_1}$, Theorem 2 implies that a GNN trained on the deterministic graph $\mathbf{G}_{n_1}$ is asymptotically transferable to any deterministic graph $\mathbf{G}_{n_2}$ in the same family where $n_2 > n_1$. This is because, as $n_1, n_2 \to \infty$, the term $\delta_c'$ converges to $\min_{i \in \mathcal{C}, j \in \{1,2\}} |\lambda_i - \lambda_{i+\text{sgn(i)}}|$, which is a fixed eigengap of $\mathbf{W}$. On the other hand, the restriction on $n_c'$ reflects a restriction on the passing band of the graph convolutions, suggesting a trade-off between the transferability and discriminability of GNNs on deterministic graphs evaluated from the same graphon.

Explicitly, in the interval $|\lambda| \in [c, 1]$, the filters $h(\lambda)$ do not have to be constant and can thus distinguish between eigenvalues. Hence, their discriminative power is larger for smaller $c$. The value of $n_c'$ counts the eigenvalues $|\lambda| \in [c, 1]$. When $c$ is small, $n_c'$ is large, which worsens the transfer error. Since the quantity $\delta_c'$ is close to the minimum graphon eigengap $\min_{i \in \mathcal{C}, j \in \{1,2\}} |\lambda_i - \lambda_{i+\text{sgn(i)}}|$ and the graphon eigenvalues accumulate near 0, $\delta_c'$ approaches 0 for small $c$. This also increases the transfer error. Therefore, small values of $c$ provide better discriminability but deteriorate transferability, and vice-versa. Still, note that for GNNs this trade-off is less rigid than for graph filters because GNNs leverage nonlinearities to scatter low frequency components ($|\lambda| \approx 0$) to larger frequencies ($|\lambda| > c$).

# 6 Numerical Results

In the following sections we illustrate the GNN transferability result on a recommendation system experiment and on the Cora dataset.

## 6.1 Movie recommendation

To illustrate Theorem 2 in a graph signal classification setting, we consider the problem of movie recommendation using the MovieLens 100k dataset (Harper and Konstan, 2016). This dataset contains 100,000 integer ratings between 1 and 5 given by $U = 943$ users to $M = 1682$ movies. Each user is seen as a node of a user similarity network, and the collection of user ratings to a given movie is a signal on this graph. To build the user network, a $U \times M$ matrix $\mathbf{R}$ is defined where $[\mathbf{R}]_{um}$ is the rating given by user $u$ to movie $m$, or 0 if this rating does not exist. The proximity between users $u_i$ and $u_j$ is then calculated as the pairwise correlation between rows $\mathbf{r}_{u_i} = [\mathbf{R}]_{u_i:}$ and $\mathbf{r}_{u_j} = [\mathbf{R}]_{u_j:}$, and each user is connected to its 40 nearest neighbors. The graph signals are the columns vectors $\mathbf{r}_m = [\mathbf{R}]_{:m}$ consisting of the user ratings to movie $m$.

Given a network with $n$ users, we implement a GNN[1] with the goal of predicting the ratings given by user 405, which is the user who has rated the most movies in the dataset (737 ratings). This GNN has $L = 1$ convolutional layer with $F = 32$ and $K = 5$, followed by a readout layer at node 405 that maps its features to a one-hot vector of dimension $C = 5$ (corresponding to ratings 1 through 5). To generate the input data, we pick the movies rated by user 405 and generate the corresponding movie signals by "zero-ing" out the ratings of user 405 while keeping the ratings given by other users. This data is then split between 90% for training and 10% for testing, with 10% of the training data used for validation. Only training data is used to build the user network in each split.

To analyze transferability, we start by training GNNs $\mathbf{\Phi}(\mathcal{H}_n; \mathbf{S}_n; \mathbf{x}_n)$ on user subnetworks consisting of random groups of $n = 100, 200, \ldots, 900$ users, including user 405. We optimize the cross-entropy loss using ADAM with learning rate $10^{-3}$ and decaying factors $\beta_1 = 0.9$ and $\beta_2 = 0.999$, and keep the models with the best validation RMSE over 40 epochs. Once the weights $\mathcal{H}_n$ are learned, we use them to define the GNN $\mathbf{\Phi}(\mathcal{H}_n; \mathbf{S}_U; \mathbf{x}_U)$, and test both $\mathbf{\Phi}(\mathcal{H}_n; \mathbf{S}_n; \mathbf{x}_n)$ and $\mathbf{\Phi}(\mathcal{H}_n; \mathbf{S}_U; \mathbf{x}_U)$ on the movies in the test set. The evolution of the difference between the test RMSEs of both GNNs, relative to the RMSE obtained on the subnetworks $\mathbf{G}_n$, is plotted in Figure 2 for 50 random splits. Note that this difference, which is proportional to $\|\mathbf{\Phi}(\mathcal{H}; \mathbf{W}_n; X_n) - \mathbf{\Phi}(\mathcal{H}; \mathbf{W}_U; X_U)\|$, decreases as the size of the subnetwork increases, following the asymptotic behavior described in Theorem 2.

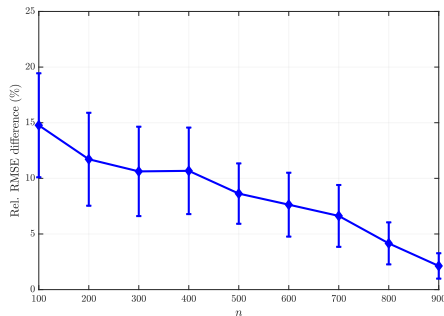

Figure 2: Relative RMSE difference on the test set for the GNNs $\Phi(\mathcal{H}; \mathbf{W}_n; X_n)$ and $\Phi(\mathcal{H}; \mathbf{W}_U; \mathbf{X}_U)$. Average over 50 random splits. Error bars have been scaled by $0.5$.

## 6.2 Cora citation network

Next, we consider a node classification experiment on the Cora dataset. The Cora dataset consists of 2708 scientific publications connected through a citation network and described by 1433 features each. The goal of the experiment is to classify these publications into 7 classes. Our base data split is the same of (Kipf and Welling, 2017), with 140 nodes for training, 500 for validation, and 1000 for testing. The remaining nodes are present in the network, but their labels are not taken into account.

To analyze transferability, we consider three scenarios in which these sets have their number of samples reduced by factors of 10, 5 and 2. The corresponding citation subgraphs have $n = 271, 542$ and 1354 nodes. In each scenario, GNNs consisting of $L = 1$ convolutional layer with $F_1 = 8$ and $K_1 = 2$, and one readout layer mapping each node's features to a one-hot vector of dimension $C = 7$, are trained using ADAM with learning rate $5 \times 10^{-3}$ and decaying factors $\beta_1 = 0.9$ and $\beta_2 = 0.999$ over 150 epochs. They are then transferred to the full 2708-node citation network to be tested on the full 1000-node test set. The average test classification accuracies on the citation subnetworks and on the full citation network are presented in Table 1 for 10, 5 and 2 random realizations of each corresponding scenario. We also report the average relative difference in accuracy with respect to the classification accuracy on the subgraphs. As $n$ increases, we see the relative accuracy difference decrease, indicating the asymptotic behavior of Theorem 2.

Table 1: Average test accuracy on subnetworks and full network for different values of $n$. Average relative accuracy difference.

| Number of nodes | $n = 271$ | $n = 542$ | $n = 1354$ |
|---|---|---|---|
| Accuracy on subnetwork $\mathbf{G}_n$ | 25.3% | 29.5% | 36.6% |
| Accuracy on full network | 27.9% | 35.4% | 41.8% |
| Relative accuracy difference | 23.1% | 22.5% | 15.4% |

## 7 Conclusions

We have introduced WNNs and shown that they can be used as generating models for GNNs on deterministic graphs evaluated from a graphon [cf. (11)]. We have also demonstrated that these GNNs can be used to approximate WNNs arbitrarily well, with an approximation error that decays asymptotically with $\mathcal{O}(n^{-0.5})$. This result was further used to prove transferability of GNNs on deterministic graphs associated with the same graphon. In particular, GNN transferabilty increases asymptotically with $\mathcal{O}(n_1^{-0.5} + n_2^{-0.5})$ for graph convolutional filters with small passing bands, suggesting a trade-off between representation power and stability for GNNs defined on deterministic graphs. GNN transferability was further demonstrated in numerical experiments of graph signal classification and node classification. Future research directions include extending the GNN transferability result to stochastic graphs sampled from graphons.

## Broader Impact

A very important implication of GNN transferability is allowing learning models to be replicated in different networks without the need for redesign. This can potentially save both data and computational resources. However, since our work utilizes standard training procedures of graph neural networks, it may inherit any potential biases present in supervised training methods, e.g. data collection bias.

## Acknowledgments and Disclosure of Funding

The work in this paper was supported by NSF CCF 1717120, ARO W911NF1710438, ARL DCIST CRA W911NF-17-2-0181, ISTC-WAS and Intel DevCloud.

## Footnotes

[1]We use the GNN library available at `https://github.com/alelab-upenn/graph-neural-networks` and implemented with PyTorch.

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
