[Supplementary Material]

# Proof of Theorem 1

To prove Theorem 1, we interpret graphon convolutions as generative models for graph convolutions. Given the graphon $\mathbf{W}(u,v) = \sum_{i \in \mathbb{Z} \setminus \{0\}} \lambda_i \varphi_i(u) \varphi_i(v)$ and a graphon convolution $Y = T_{\mathbf{H}} X$ written as

$$(T_{\mathbf{H}} X)(v) = \sum_{i \in \mathbb{Z} \setminus \{0\}} h(\lambda_i) \varphi_i(v) \int_0^1 \varphi_i(u) X(u) du \qquad (1)$$

we can generate graph convolutions $\mathbf{y}_n = \mathbf{H}_n(\mathbf{S}_n)\mathbf{x}_n$ by defining $u_i = (i-1)/n$ for $1 \le i \le n$ and setting

$$\begin{aligned} [\mathbf{S}_n]_{ij} &= \mathbf{W}(u_i, u_j) \\ [\mathbf{x}_n]_i &= X(u_i) \\ \mathbf{H}_n(\mathbf{S}_n)\mathbf{x}_n &= \mathbf{V}_n^{\mathsf{H}} h(\mathbf{\Lambda}_n) \mathbf{V}_n^{\mathsf{H}} \mathbf{x}_n \end{aligned} \qquad (2)$$

where $\mathbf{S}_n$ is the GSO of $\mathbf{G}_n$, the deterministic graph obtained from $\mathbf{W}$ as in Section 3.2.1, $\mathbf{x}_n$ is the deterministic graph signal obtained by evaluating the graphon signal $X$ at points $u_i$, and $\mathbf{\Lambda}_n$ and $\mathbf{V}_n$ are the eigenvalues and eigenvectors of $\mathbf{S}_n$ respectively. It is also possible to define graphon convolutions induced by graph convolutions. The graph convolution $\mathbf{y}_n = \mathbf{H}_n(\mathbf{S}_n)\mathbf{x}_n$ induces a graphon convolution $Y_n = T_{\mathbf{H}_n} X_n$ obtained by constructing a partition $I_1 \cup \ldots \cup I_n$ of $[0,1]$ with $I_i = [(i-1)/n, i/n]$ and defining

$$\begin{aligned} \mathbf{W}_n(u,v) &= [\mathbf{S}_n]_{ij} \times \mathbb{I}(u \in I_i)\mathbb{I}(v \in I_j) \\ X_n(u) &= [\mathbf{x}_n]_i \times \mathbb{I}(u \in I_i) \\ (T_{\mathbf{H}_n} X_n)(v) &= \sum_{i \in \mathbb{Z} \setminus \{0\}} h(\lambda_i^n) \varphi_i^n(v) \int_0^1 \varphi_i^n(u) X_n(u) du \end{aligned} \qquad (3)$$

where $\mathbf{W}_n$ is the *graphon induced by* $\mathbf{G}_n$, $X_n$ is the graphon signal induced by the graph signal $\mathbf{x}_n$ and $\lambda_i^n$ and $\varphi_i^n$ are the eigenvalues and eigenfunctions of $\mathbf{W}_n$.

Theorem 1 is a direct consequence of the following theorem, which states that graphon convolutions can be approximated by graph convolutions on large graphs.

**AS1.** *The graphon $\mathbf{W}$ is $A_1$-Lipschitz, i.e.* $|\mathbf{W}(u_2, v_2) - \mathbf{W}(u_1, v_1)| \le A_1(|u_2 - u_1| + |v_2 - v_1|)$.

**AS2.** *The convolutional filters $h$ are $A_2$-Lipschitz and non-amplifying, i.e.* $|h(\lambda)| < 1$.

**AS3.** *The graphon signal $X$ is $A_3$-Lipschitz.*

**Theorem 3.** Consider the graphon convolution given by $Y = T_{\mathbf{H}} X$ as in (1), where $h(\lambda)$ is constant for $|\lambda| < c$. For the graph convolution instantiated from $T_{\mathbf{H}}$ as $\mathbf{y}_n = \mathbf{H}_n(\mathbf{S}_n)\mathbf{x}_n$ [cf. (2)], under Assumptions 1 through 3 it holds

$$\|Y - Y_n\|_{L_2} \le \sqrt{A_1} \left( A_2 + \frac{\pi n_c}{\delta_c} \right) n^{-\frac{1}{2}} \|X\|_{L_2} + \frac{2A_3}{\sqrt{3}} n^{-\frac{1}{2}} \qquad (4)$$

where $Y_n = T_{\mathbf{H}_n} X_n$ is the graph convolution induced by $\mathbf{y}_n = \mathbf{H}_n(\mathbf{S}_n)\mathbf{x}_n$ [cf. (3)], $n_c$ is the cardinality of the set $\mathcal{C} = \{i \mid |\lambda_i^n| \ge c\}$, and $\delta_c = \min_{i \in \mathcal{C}}(|\lambda_i - \lambda_{i+\mathrm{sgn}(i)}^n|, |\lambda_{i+\mathrm{sgn}(i)} - \lambda_i^n|, |\lambda_1 - \lambda_{-1}^n|, |\lambda_1^n - \lambda_{-1}|)$, with $\lambda_i$ and $\lambda_i^n$ denoting the eigenvalues of $\mathbf{W}$ and $\mathbf{W}_n$ respectively. In particular, if $X = X_n$ we have

$$\|Y - Y_n\|_{L_2} \le \sqrt{A_1} \left( A_2 + \frac{\pi n_c}{\delta_c} \right) n^{-\frac{1}{2}} \|X\|_{L_2}. \qquad (5)$$

*Proof of Theorem 3.* To prove Theorem 3, we need the following four propositions.

**Proposition 1.** *Let $\mathbf{W} : [0,1]^2 \to [0,1]$ be an $A_1$-Lipschitz graphon, and let $\mathbf{W}_n$ be the graphon induced by the deterministic graph $\mathbf{G}_n$ obtained from $\mathbf{W}$ as in Section 3.2.1. The $L_2$ norm of $\mathbf{W} - \mathbf{W}_n$ satisfies*

$$\|\mathbf{W} - \mathbf{W}_n\|_{L_2([0,1]^2)} \le \sqrt{\|\mathbf{W} - \mathbf{W}_n\|_{L_1([0,1]^2)}} \le \frac{\sqrt{A_1}}{\sqrt{n}}.$$

*Proof.* Partitioning the unit interval as $I_i = [(i-1)/n, i/n]$ for $1 \le i \le n$ (the same partition used to obtain $\mathbf{S}_n$, and thus $\mathbf{W}_n$, from $\mathbf{W}$), we can use the graphon's Lipschitz property to derive

$$\|\mathbf{W} - \mathbf{W}_n\|_{L_1(I_i \times I_j)} \le A_1 \int_0^{1/n} \int_0^{1/n} |u| du dv + A_1 \int_0^{1/n} \int_0^{1/n} |v| dv du = \frac{A_1}{2n^3} + \frac{A_1}{2n^3} = \frac{A_1}{n^3}.$$

We can then write

$$\|\mathbf{W} - \mathbf{W}_n\|_{L_1([0,1]^2)} = \sum_{i,j} \|\mathbf{W} - \mathbf{W}_n\|_{L_1(I_i \times I_j)} \le n^2 \frac{A_1}{n^3} = \frac{A_1}{n}$$

which, since $\mathbf{W} - \mathbf{W}_n : [0,1]^2 \to [-1,1]$, implies

$$\|\mathbf{W} - \mathbf{W}_n\|_{L_2([0,1]^2)} \le \sqrt{\|\mathbf{W} - \mathbf{W}_n\|_{L_1([0,1]^2)}} \le \frac{\sqrt{A_1}}{\sqrt{n}}.$$

$\square$

**Proposition 2.** *Let $T$ and $T'$ be two self-adjoint operators on a separable Hilbert space $\mathcal{H}$ whose spectra are partitioned as $\gamma \cup \Gamma$ and $\omega \cup \Omega$ respectively, with $\gamma \cap \Gamma = \emptyset$ and $\omega \cap \Omega = \emptyset$. If there exists $d > 0$ such that $\min_{x \in \gamma, \, y \in \Omega} |x - y| \ge d$ and $\min_{x \in \omega, \, y \in \Gamma} |x - y| \ge d$, then*

$$\|E_T(\gamma) - E_{T'}(\omega)\| \le \frac{\pi}{2} \frac{\|T - T'\|}{d}$$

*Proof.* See (Seelmann, 2014). $\square$

**Proposition 3.** *Let $X \in L_2([0,1])$ be an $A_3$-Lipschitz graphon signal, and let $X_n$ be the graphon signal induced by the deterministic graph signal $\mathbf{x}_n$ obtained from $X$ as in (2). The $L_2$ norm of $X - X_n$ satisfies*

$$\|X - X_n\|_{L_2([0,1])} \le \frac{A_3}{\sqrt{3n}}.$$

*Proof.* Partitioning the unit interval as $I_i = [(i-1)/n, i/n]$ for $1 \le i \le n$ (the same partition used to obtain $\mathbf{x}_n$, and thus $X_n$, from $X$), we can use the Lipschitz property of $X$ to derive

$$\|X - X_n\|_{L_2(I_i)} \le \sqrt{A_3^2 \int_0^{1/n} u^2 du} = \sqrt{\frac{A_3^2}{3n^3}} + \frac{A_3}{n\sqrt{3n}}.$$

We can then write

$$\|X - X_n\|_{L_2([0,1])} = \sum_i \|X - X_n\|_{L_2(I_i)} \le n \frac{A_3}{n\sqrt{3n}} = \frac{A_3}{\sqrt{3n}}.$$

$\square$

**Proposition 4.** *Let $\mathbf{W} : [0,1]^2 \to [0,1]$ and $\mathbf{W}' : [0,1]^2 \to [0,1]$ be two graphons with eigenvalues given by $\{\lambda_i(T_\mathbf{W})\}_{i \in \mathbb{Z} \setminus \{0\}}$ and $\{\lambda_i(T_{\mathbf{W}'})\}_{i \in \mathbb{Z} \setminus \{0\}}$, ordered according to their sign and in decreasing order of absolute value. Then, for all $i \in \mathbb{Z} \setminus \{0\}$, the following inequalities hold*

$$|\lambda_i(T_{\mathbf{W}'}) - \lambda_i(T_\mathbf{W})| \le \|T_{\mathbf{W}' - \mathbf{W}}\| \le \|\mathbf{W}' - \mathbf{W}\|_{L_2}.$$

*Proof.* Let $\mathbf{A} := \mathbf{W}' - \mathbf{W}$ and let $S_k$ denote a $k$-dimensional subspace of $L_2([0,1])$. Using the minimax principle (Kato, 2013, Chapter 1.6.10), we can write

$$\lambda_k(T_\mathbf{W}) = \min_{S_{k-1}} \max_{X \in S_{k-1}^\perp, \|X\|_{L_2} = 1} \langle T_\mathbf{W} X, X \rangle.$$

Therefore, it holds that

$$\lambda_k(T_{\mathbf{W}'}) = \lambda_k(T_{\mathbf{W}+\mathbf{A}}) = \min_{S_{k-1}} \max_{X \in S_{k-1}^\perp, \|X\|_{L_2}=1} \langle T_{\mathbf{W}+\mathbf{A}} X, X \rangle = \min_{S_{k-1}} \max_{X \in S_{k-1}^\perp, \|X\|_{L_2}=1} \langle T_{\mathbf{W}} + T_{\mathbf{A}} X, X \rangle$$

$$= \min_{S_{k-1}} \max_{X \in S_{k-1}^\perp, \|X\|_{L_2}=1} (\langle T_{\mathbf{W}} X, X \rangle + \langle T_{\mathbf{A}} X, X \rangle)$$

$$\leq \min_{S_{k-1}} \left( \max_{X \in S_{k-1}^\perp, \|X\|_{L_2}=1} \langle T_{\mathbf{W}} X, X \rangle + \max_{X \in S_{k-1}^\perp, \|X\|_{L_2}=1} \langle T_{\mathbf{A}} X, X \rangle \right)$$

$$\leq \min_{S_{k-1}} \left( \max_{X \in S_{k-1}^\perp, \|X\|_{L_2}=1} \langle T_{\mathbf{W}} X, X \rangle + \max_\ell \lambda_\ell(T_{\mathbf{A}}) \right)$$

$$= \min_{S_{k-1}} \max_{X \in S_{k-1}^\perp, \|X\|_{L_2}=1} \langle T_{\mathbf{W}} X, X \rangle + \max_\ell \lambda_\ell(T_{\mathbf{A}}) = \lambda_k(T_{\mathbf{W}}) + \max_\ell \lambda_\ell(T_{\mathbf{A}}) .$$

where the first inequality follows from $\max(a+b) \leq \max(a) + \max(b)$ and the second from the fact that $\langle T_{\mathbf{A}} X, X \rangle \leq \max_\ell \lambda_\ell(T_{\mathbf{A}})$ for all unitary $X$. Rearranging terms and using the definition of the operator norm, we get

$$\lambda_k(T_{\mathbf{W}'}) - \lambda_k(T_{\mathbf{W}}) \leq \max_\ell \lambda_\ell(T_{\mathbf{A}}) \leq \max_\ell |\lambda_\ell(T_{\mathbf{A}})| = \|T_{\mathbf{A}}\| \leq \|A\|_{L_2}. \tag{6}$$

where we have also used the fact that the Hilbert-Schmidt norm dominates the operator norm.

To prove that this inequality holds in absolute value, let $\mathbf{A}' = -\mathbf{A}$. Following the same reasoning as before, we get

$$\lambda_k(T_{\mathbf{W}}) = \lambda_k(T_{\mathbf{W}'+A'}) \leq \lambda_k(T_{\mathbf{W}'}) + \|T_{\mathbf{A}'}\| \leq \lambda_k(T_{\mathbf{W}'}) + \|A'\|_{L_2}$$

and since $\|T_{\mathbf{A}'}\| = \|T_{\mathbf{A}}\|$ and $\|A'\|_{L_2} = \|A\|_{L_2}$,

$$\lambda_k(T_{\mathbf{W}}) - \lambda_k(T_{\mathbf{W}'}) \leq \|T_{\mathbf{A}}\| \leq \|A\|_{L_2} . \tag{7}$$

Putting (6) and (7) together completes the proof. $\square$

We first prove the result of Theorem 3 for filters $h(\lambda)$ satisfying $h(\lambda) = 0$ for $|\lambda| < c$. Using the triangle inequality, we can write the norm difference $\|Y - Y_n\|_{L_2}$ as

$$\|Y - Y_n\|_{L_2} = \|T_{\mathbf{H}} X - T_{\mathbf{H}_n} X_n\|_{L_2} = \|T_{\mathbf{H}} X + T_{\mathbf{H}_n} X - T_{\mathbf{H}_n} X - T_{\mathbf{H}_n} X_n\|_{L_2}$$

$$\leq \|T_{\mathbf{H}} X - T_{\mathbf{H}_n} X\|_{L_2} \textbf{(1)} + \|T_{\mathbf{H}_n} (X - X_n)\|_{L_2} \textbf{(2)}$$

where the LHS is split between terms **(1)** and **(2)**.

Writing the inner products $\int_0^1 X(u)\varphi_i(u)du$ and $\int_0^1 X(u)\varphi_i^n(u)du$ as $\hat{X}(\lambda_i)$ and $\hat{X}(\lambda_i^n)$ for simplicity, we can then express **(1)** as

$$\|T_{\mathbf{H}} X - T_{\mathbf{H}_n} X\|_{L_2} = \left\| \sum_i h(\lambda_i)\hat{X}(\lambda_i)\varphi_i - \sum_i h(\lambda_i^n)\hat{X}(\lambda_i^n)\varphi_i^n \right\|_{L_2}$$

$$= \left\| \sum_i h(\lambda_i)\hat{X}(\lambda_i)\varphi_i - h(\lambda_i^n)\hat{X}(\lambda_i^n)\varphi_i^n \right\|_{L_2} .$$

Using the triangle inequality, this becomes

$$\|T_{\mathbf{H}} X - T_{\mathbf{H}_n} X\|_{L_2} = \left\| \sum_i h(\lambda_i)\hat{X}(\lambda_i)\varphi_i - h(\lambda_i^n)\hat{X}(\lambda_i^n)\varphi_i^n \right\|_{L_2}$$

$$= \left\| \sum_i h(\lambda_i)\hat{X}(\lambda_i)\varphi_i + h(\lambda_i^n)\hat{X}(\lambda_i)\varphi_i - h(\lambda_i^n)\hat{X}(\lambda_i)\varphi_i - h(\lambda_i^n)\hat{X}(\lambda_i^n)\varphi_i^n \right\|_{L_2}$$

$$\leq \left\| \sum_i (h(\lambda_i) - h(\lambda_i^n)) \hat{X}(\lambda_i)\varphi_i \right\|_{L_2} \textbf{(1.1)}$$

$$+ \left\| \sum_i h(\lambda_i^n) \left( \hat{X}(\lambda_i)\varphi_i - \hat{X}(\lambda_i^n)\varphi_i^n \right) \right\|_{L_2} \textbf{(1.2)}$$

where we have now split **(1)** between **(1.1)** and **(1.2)**.

Focusing on **(1.1)**, note that the filter's Lipschitz property allows writing $h(\lambda_i) - h(\lambda_i^n) \leq A_2 |\lambda_i - \lambda_i^n|$. Hence, using Proposition 4 together with the Cauchy-Schwarz inequality, we get

$$\left\| \sum_i \left( h(\lambda_i) - h(\lambda_i^n) \right) \hat{X}(\lambda_i) \varphi_i \right\|_{L_2} \leq A_2 \|\mathbf{W} - \mathbf{W}_n\|_{L_2} \left\| \sum_i \hat{X}(\lambda_i) \varphi_i \right\|_{L_2}$$

and, from Proposition 1,

$$\left\| \sum_i \left( h(\lambda_i) - h(\lambda_i^n) \right) \hat{X}(\lambda_i) \varphi_i \right\|_{L_2} \leq \frac{A_2 \sqrt{A_1}}{\sqrt{n}} \|X\|_{L_2}. \tag{8}$$

For **(1.2)**, we use the triangle and Cauchy-Schwarz inequalities to write

$$\left\| \sum_i h(\lambda_i^n) \left( \hat{X}(\lambda_i) \varphi_i - \hat{X}(\lambda_i^n) \varphi_i^n \right) \right\|_{L_2} = \left\| \sum_i h(\lambda_i^n) \left( \hat{X}(\lambda_i) \varphi_i + \hat{X}(\lambda_i) \varphi_i^n - \hat{X}(\lambda_i) \varphi_i^n - \hat{X}(\lambda_i^n) \varphi_i^n \right) \right\|_{L_2}$$

$$\leq \left\| \sum_i h(\lambda_i^n) \hat{X}(\lambda_i) (\varphi_i - \varphi_i^n) \right\|_{L_2} + \left\| \sum_i h(\lambda_i^n) \varphi_i^n \langle X, \varphi_i - \varphi_i^n \rangle \right\|_{L_2}$$

$$\leq 2 \sum_i \|h(\lambda_i^n)\|_{L_2} \|X\|_{L_2} \|\varphi_i - \varphi_i^n\|_{L_2}.$$

Using Proposition 2 with $\gamma = \lambda_i$ and $\omega = \lambda_i^n$, we then get

$$\left\| \sum_i h(\lambda_i^n) \left( \hat{X}(\lambda_i) \varphi_i - \hat{X}(\lambda_i^n) \varphi_i^n \right) \right\|_{L_2} \leq \|X\|_{L_2} \sum_i \|h(\lambda_i^n)\|_{L_2} \frac{\pi \|T_{\mathbf{W}} - T_{\mathbf{W}_n}\|}{d_i}$$

where $d_i$ is the minimum between $\min(|\lambda_i - \lambda_{i+1}^n|, |\lambda_i - \lambda_{i-1}^n|)$ and $\min(|\lambda_i^n - \lambda_{i+1}|, |\lambda_i^n - \lambda_{i-1}|)$ for each $i$. Since $\delta_c \leq d_i$ for all $i$ and $\|T_{\mathbf{W}} - T_{\mathbf{W}_n}\| \leq \|\mathbf{W} - \mathbf{W}_n\|_{L_2}$ (i.e., the Hilbert-Schmidt norm dominates the operator norm), this becomes

$$\left\| \sum_i h(\lambda_i^n) \left( \hat{X}(\lambda_i) \varphi_i - \hat{X}(\lambda_i^n) \varphi_i^n \right) \right\|_{L_2} \leq \frac{\pi \|\mathbf{W} - \mathbf{W}_n\|_{L_2}}{\delta_c} \|X\|_{L_2} \sum_i \|h(\lambda_i^n)\|_{L_2}$$

and, using Proposition 1,

$$\left\| \sum_i h(\lambda_i^n) \left( \hat{X}(\lambda_i) \varphi_i - \hat{X}(\lambda_i^n) \varphi_i^n \right) \right\|_{L_2} \leq \frac{\pi \sqrt{A_1}}{\delta_c \sqrt{n}} \|X\|_{L_2} \sum_i \|h(\lambda_i^n)\|_{L_2}.$$

The final bound for **(1.2)** is obtained by noting that $|h(\lambda)| < 1$ and $h(\lambda) = 0$ for $|\lambda| < c$. Since there are a total of $n_c$ eigenvalues $\lambda_i^n$ for which $|\lambda_i^n| \geq c$, we get

$$\left\| \sum_i h(\lambda_i^n) \left( \hat{X}(\lambda_i) \varphi_i - \hat{X}(\lambda_i^n) \varphi_i^n \right) \right\|_{L_2} \leq \frac{\pi \sqrt{A_1}}{\delta_c \sqrt{n}} \|X\|_{L_2} n_c. \tag{9}$$

A bound for **(2)** follows immediately from Proposition 3. Since $|h(\lambda)| < 1$, the norm of the operator $T_{\mathbf{H}_n}$ is bounded by 1. Using the Cauchy-Schwarz inequality, we then have $\|T_{\mathbf{H}_n}(X - X_n)\|_{L_2} \leq \|X - X_n\|_{L_2}$ and therefore

$$\|T_{\mathbf{H}_n}(X - X_n)\|_{L_2} \leq \frac{A_3}{\sqrt{3n}} \tag{10}$$

which completes the bound on $\|Y - Y_n\|_{L_2}$ when $h(\lambda) = 0$ for $|\lambda| < c$. For filters in which $h(\lambda)$ is a constant for $\lambda < c$, we obtain a bound by observing that $h(\lambda)$ can be constructed as the sum of two filters: an $A_2$-Lipschitz filter $f(\lambda)$ with $f(\lambda) = 0$ for $|\lambda| < c$, and a bandpass filter $g(\lambda)$ with $g(\lambda)$ constant for $|\lambda| < c$ and 0 otherwise. Hence, by the triangle inequality

$$\|Y - Y_n\|_{L_2} = \|T_{\mathbf{H}} X - T_{\mathbf{H}_n}\|_{L_2} \leq \|T_{\mathbf{F}} X - T_{\mathbf{F}_n} X_n\|_{L_2} + \|T_{\mathbf{G}} X - T_{\mathbf{G}_n} X_n\|_{L_2}.$$

The bound on $\|T_{\mathbf{F}}X - T_{\mathbf{F}_n}\|_{L_2}$ is the one we have derived, and for $\|T_{\mathbf{G}}X - T_{\mathbf{G}_n}X_n\|_{L_2}$, we use $|g(\lambda)| \leq 1$ and the fact that $g(\lambda)$ is constant in $[0, c]$ with $0 < c \leq 1$ to obtain

$$\|T_{\mathbf{G}}X - T_{\mathbf{G}_n}X_n\|_{L_2} \leq \|X - X_n\|_{L_2} \leq \frac{A_3}{\sqrt{3n}} \tag{11}$$

where the last inequality follows from Proposition 3.

Putting together (8), (9), (10) and (11), we arrive at the first result of the theorem as stated in (4). The second result [cf. (5)] is obtained by observing that, for $X = X_n$, bound (2) in (10) simplifies to $\|T_{\mathbf{H}_n}(X - X_n)\|_{L_2} = 0$; and, similarly in (11), $\|T_{\mathbf{G}}X - T_{\mathbf{G}_n}X_n\|_{L_2} \leq \|X - X_n\|_{L_2} = 0$. $\quad\square$

*Proof of Theorem 1.* To compute a bound for $\|Y - Y_n\|_{L_2}$, we start by writing it in terms of the last layer's features as

$$\|Y - Y_n\|_{L_2}^2 = \sum_{f=1}^{F_L} \left\|X_L^f - X_{n,L}^f\right\|_{L_2}^2. \tag{12}$$

At layer $\ell$ of the WNN $\boldsymbol{\Phi}(\mathcal{H}; \mathbf{W}; X)$, we have

$$X_\ell^f = \rho\left(\sum_{g=1}^{F_{\ell-1}} \mathbf{h}_\ell^{fg} *_{\mathbf{W}} X_{\ell-1}^g\right) = \rho\left(\sum_{g=1}^{F_{\ell-1}} T_{\mathbf{H}_\ell^{fg}} X_{\ell-1}^g\right)$$

and similarly for $\boldsymbol{\Phi}(\mathcal{H}; \mathbf{W}_n; X_n)$,

$$X_{n,\ell}^f = \rho\left(\sum_{g=1}^{F_{\ell-1}} \mathbf{h}_{n,\ell}^{fg} *_{\mathbf{W}} X_{n,\ell-1}^g\right) = \rho\left(\sum_{g=1}^{F_{\ell-1}} T_{\mathbf{H}_{n,\ell}^{fg}} X_{n,\ell-1}^g\right).$$

We can therefore write $\|X_\ell^f - X_{n,\ell}^f\|_{L_2}$ as

$$\left\|X_\ell^f - X_{n,\ell}^f\right\|_{L_2} = \left\|\rho\left(\sum_{g=1}^{F_{\ell-1}} T_{\mathbf{H}_\ell^{fg}} X_{\ell-1}^g\right) - \rho\left(\sum_{g=1}^{F_{\ell-1}} T_{\mathbf{H}_{n,\ell}^{fg}} X_{n,\ell-1}^g\right)\right\|_{L_2}$$

and, since $\rho$ is normalized Lipschitz,

$$\left\|X_\ell^f - X_{n,\ell}^f\right\|_{L_2} \leq \left\|\sum_{g=1}^{F_{\ell-1}} T_{\mathbf{H}_\ell^{fg}} X_{\ell-1}^g - T_{\mathbf{H}_{n,\ell}^{fg}} X_{n,\ell-1}^g\right\|_{L_2}$$

$$\leq \sum_{g=1}^{F_{\ell-1}} \left\|T_{\mathbf{H}_\ell^{fg}} X_{\ell-1}^g - T_{\mathbf{H}_{n,\ell}^{fg}} X_{n,\ell-1}^g\right\|_{L_2}.$$

where the second inequality follows from the triangle inequality. Looking at each feature $g$ independently, we apply the triangle inequality once again to get

$$\left\|T_{\mathbf{H}_\ell^{fg}} X_{\ell-1}^g - T_{\mathbf{H}_{n,\ell}^{fg}} X_{n,\ell-1}^g\right\|_{L_2} \leq \left\|T_{\mathbf{H}_\ell^{fg}} X_{\ell-1}^g - T_{\mathbf{H}_{n,\ell}^{fg}} X_{\ell-1}^g\right\|_{L_2} + \left\|T_{\mathbf{H}_{n,\ell}^{fg}}\left(X_{\ell-1}^g - X_{n,\ell-1}^g\right)\right\|_{L_2}.$$

The first term on the RHS of this inequality is bounded by (5) in Theorem 3. The second term can be decomposed by using Cauchy-Schwarz and recalling that $|h(\lambda)| < 1$ for all graphon convolutions in the WNN (Assumption 1). We thus obtain a recursion for $\|X_\ell^f - X_{n,\ell}^f\|_{L_2}$, which is given by

$$\left\|X_\ell^f - X_{n,\ell}^f\right\|_{L_2} \leq \sum_{g=1}^{F_{\ell-1}} \sqrt{A_1}\left(A_2 + \frac{\pi n_c}{\delta_c}\right) n^{-\frac{1}{2}} \|X_{\ell-1}^g\|_{L_2} + \sum_{g=1}^{F_{\ell-1}} \left\|X_{\ell-1}^g - X_{n,\ell-1}^g\right\|_{L_2} \tag{13}$$

and whose first term, $\sum_{g=1}^{F_0} \|X_0^g - X_{n,0}^g\|_{L_2} = \sum_{g=1}^{F_0} \|X^g - X_n^g\|_{L_2}$, is bounded as $\sum_{g=1}^{F_0} \|X_0^g - X_{n,0}^g\|_{L_2} \leq F_0 A_3/\sqrt{3n}$ by Proposition 3.

To solve this recursion, we need to compute the norm $\|X_{\ell-1}^g\|_{L_2}$. Since the nonlinearity $\rho$ is normalized Lipschitz and $\rho(0) = 0$ by Assumption 2, this bound can be written as

$$\left\|X_{\ell-1}^g\right\|_{L_2} \leq \left\|\sum_{g=1}^{F_{\ell-1}} T_{\mathbf{H}_\ell^{fg}} X_{\ell-1}^g\right\|_{L_2}$$

and using the triangle and Cauchy Schwarz inequalities,

$$\left\|X_{\ell-1}^g\right\|_{L_2} \leq \sum_{g=1}^{F_{\ell-1}} \left\|T_{\mathbf{H}_\ell^{fg}}\right\|_{L_2} \left\|X_{\ell-1}^g\right\|_{L_2} \leq \sum_{g=1}^{F_{\ell-1}} \left\|X_{\ell-1}^g\right\|_{L_2}$$

where the second inequality follows from $|h(\lambda)| < 1$. Expanding this expression with initial condition $X_0^g = X^g$ yields

$$\left\|X_{\ell-1}^g\right\|_{L_2} \leq \prod_{\ell'=1}^{\ell-1} F_{\ell'} \sum_{g=1}^{F_0} \|X^g\|_{L_2}. \tag{14}$$

and substituting it back in (13) to solve the recursion, we get

$$\left\|X_\ell^f - X_{n,\ell}^f\right\|_{L_2} \leq L\sqrt{A_1}\left(A_2 + \frac{\pi n_c}{\delta_c}\right) n^{-\frac{1}{2}} \left(\prod_{\ell'=1}^{\ell-1} F_{\ell'}\right) \sum_{g=1}^{F_0} \|X^g\|_{L_2} + \frac{F_0 A_3}{\sqrt{3}} n^{-\frac{1}{2}}. \tag{15}$$

To arrive at the result of Theorem 1, we evaluate (15) with $\ell = L$ and substitute it into (12) to obtain

$$\begin{aligned}
\|Y - Y_n\|_{L_2}^2 &= \sum_{f=1}^{F_L} \left\|X_L^f - X_{n,L}^f\right\|_{L_2}^2 \\
&\leq \sum_{f=1}^{F_L} \left(L\sqrt{A_1}\left(A_2 + \frac{\pi n_c}{\delta_c}\right) n^{-\frac{1}{2}} \left(\prod_{\ell=1}^{L-1} F_\ell\right) \sum_{g=1}^{F_0} \|X^g\|_{L_2} + \frac{F_0 A_3}{\sqrt{3}} n^{-\frac{1}{2}}\right)^2.
\end{aligned} \tag{16}$$

Finally, since $F_0 = F_L = 1$ and $F_\ell = F$ for $1 \leq \ell \leq L-1$,

$$\|Y - Y_n\|_{L_2} \leq L\sqrt{A_1}\left(A_2 + \frac{\pi n_c}{\delta_c}\right) n^{-\frac{1}{2}} F^{L-1} \|X\|_{L_2} + \frac{A_3}{\sqrt{3}} n^{-\frac{1}{2}}. \tag{17}$$

$\square$

## Proof of Theorem 2

Theorem 2 follows directly from Theorem 1 via the triangle inequality.

*Proof of Theorem 2.* By the triangle inequality, we can bound $\|Y_{n_1} - Y_{n_2}\|_{L_2}$ as

$$\|Y_{n_1} - Y_{n_2}\|_{L_2} = \|Y_{n_1} - Y + Y - Y_{n_2}\|_{L_2} \leq \|Y_{n_1} - Y\|_{L_2} + \|Y - Y_{n_2}\|_{L_2}.$$

Theorem 1 gives a bound for both $\|Y_{n_1} - Y\|_{L_2}$ and $\|Y - Y_{n_2}\|_{L_2}$. Setting $n_c' = \max_{j \in \{1,2\}} |\mathcal{C}_j|$, $\mathcal{C}_j = \{i \mid |\lambda_i^{n_j}| \geq c\}$, and $\delta_c' = \min_{i \in \mathcal{C}_j, j \in \{1,2\}}(|\lambda_i - \lambda_{i+\mathrm{sgn}(i)}^{n_j}|, |\lambda_{i+\mathrm{sgn}(i)} - \lambda_i^{n_j}|, |\lambda_1 - \lambda_{-1}^{n_j}|, |\lambda_1^{n_j} - \lambda_{-1}|)$, we arrive at the theorem's result. $\square$

## Additional Numerical Results: Consensus

In this section, we provide a second set of experiments to illustrate the effect of the hyperparameters $F$, $K$, and $L$ (number of features, filter taps, and layers respectively) on GNN transferability. These experiments are based on the consensus problem, in which the goal is to drive the signal values at each node to the average of the graph signal over all nodes.

(a) $K = 4$, $L = 1$      (b) $F = 8$, $L = 1$      (c) $F = 8$, $K = 4$

Figure 1: Relative difference between test rRMSEs achieved on the graphs $\mathbf{G}_n$ and $\mathbf{G}_N$ ($N = 2000$) for different configurations of $F$, $K$ and $L$. Average and standard deviation over 3 graph realizations and 3 data realizations per graph. Error bars have been scaled by 1.5. (a) Fixed $K$ and $L$, varying number of features $F$. (b) Fixed $F$ and $L$, varying number of filter taps $K$. (c) Fixed $F$ and $K$, varying number of layers $L$.

The problem setup is as follows. Given a network $\mathbf{G}_n$, the input data $\mathbf{x}_n$ is generated by sampling a graph signal $\mathbf{x}_n = |\tilde{\mathbf{x}}_n|$ from a folded multivariate normal distribution with mean $\tilde{\boldsymbol{\mu}}$ and covariance $\tilde{\boldsymbol{\Sigma}}$. Explicitly, $\tilde{\mathbf{x}}_n \sim \mathcal{N}(\tilde{\boldsymbol{\mu}}, \tilde{\boldsymbol{\Sigma}})$, where we set $\tilde{\boldsymbol{\mu}} = \mathbf{0}$ and $\tilde{\boldsymbol{\Sigma}} = 100\mathbf{I}$. The output data is given by

$$\mathbf{y}_n = \frac{\sum_{i=1}^n [\mathbf{x}_n]_i}{n} \mathbf{1}$$

which is also a graph signal on $\mathbf{G}_n$. This data is split between 8400 input-output pairs for training, 200 for validation, and 200 for testing.

To analyze transferability, we train GNNs $\boldsymbol{\Phi}(\mathcal{H}_n; \mathbf{S}_n; \mathbf{x}_n)$ on small networks of size $n$ for multiple values of $n$, and use the learned parameter sets $\mathcal{H}_n$ to define and test GNNs $\boldsymbol{\Phi}(\mathcal{H}_n; \mathbf{S}_N; \mathbf{x}_N)$ on a network of size $N \gg n$. The networks are all stochastic block model graphs with 2 (balanced) communities, intra-community probability $p_{c_i c_i} = 0.8$ and inter-community probability $p_{c_i c_j} = 0.2$. We set $N = 2000$ and vary $n$ as $n = 50, 250, 500, 1000$.

The GNN parameters are learned by optimizing the L1 loss on the training set using ADAM with learning rate 0.001 and decay factors $\beta_1 = 0.9$ and $\beta_2 = 0.999$. We keep the model with smallest relative RMSE (rRMSE) on the validation set over 40 epochs. The performance metric for transferability is the relative difference between the test rRMSEs achieved by the GNN on $\mathbf{G}_n$ and the GNN on $\mathbf{G}_N$, i.e., the difference between the rRMSEs obtained by $\boldsymbol{\Phi}(\mathcal{H}_n; \mathbf{S}_n; \mathbf{x}_n)$ and $\boldsymbol{\Phi}(\mathcal{H}_n; \mathbf{S}_N; \mathbf{x}_N)$ on the test set relative to the test rRMSE for $\boldsymbol{\Phi}(\mathcal{H}_n; \mathbf{S}_n; \mathbf{x}_n)$. This relative rRMSE difference is reported in Figures 1a through 1c for 3 graph realizations and 3 data realizations per graph, as well as different values of $F$, $K$, and $L$. In Figure 1a, $K$ and $L$ are fixed at $K = 4$ and $L = 1$, and we vary $F$. In Figure 1b, $F$ and $L$ are fixed at $F = 8$ and $L = 1$, and we vary $K$. In Figure 1c, we fix $F = 8$ and $K = 4$ and vary the number of layers $L$.

Note that, for all configurations of $F$, $K$ and $L$, the average relative rRMSE difference and its standard deviation decrease as $n$ increases, agreeing with Theorem 2. In Figure 1a, we can also see that for smaller values of $n$ the error bars are smaller for $F = 4$ than they are for $F = 8$ and $F = 16$. This illustrates the dependence of the transferability bound on the GNN width. In Figure 1b, varying the number of filter taps $K$ does not seem to have much of an effect on GNN transferability. In contrast, in Figure 1c we can clearly see the size of the error bars increase with the number of layers $L$, especially for small $n$. This is expected, since $L$ exponentiates $F$ in the transferability bound of Theorem 2.