[Reviews · NeurIPS 2020]

Review 1

Summary and Contributions: This paper addresses the research question: To what extent is a GNN prediction stable when the underlying graph changes? The main theorem shows that when we apply a common GNN to two different graphs with $n_1$ and $n_2$ nodes obtained deterministically from the same graphon, the output differences are $O(n_1^{-1/2} + n_2^{-1/2})$ in terms of the L2 norm. In this sense, GNNs have transferability. Also, the theorem suggests that there is a trade-off between transferability and discriminability.

Strengths: Soundness of the claims - The theoretical results (Theorem1, 2) properly formulate the main claim that GNN has transferability between different graphs sampled from the same graphon. It is also appropriate to say that the obtained upper bound has a trade-off between "transferability" and "discriminability." Regarding the trade-off, the current version of the paper explains it only briefly in l.266--268. I think it is better to explain how we can derive the trade-off from the upper bound in more detail. Significance and novelty of the contribution - There is little theoretical research about the transferability of GNNs. The only study that I know is [Keriven et al., 20], as I describe in the later section). Relevance to the NeurIPS community. - The research of GNNs is one of the main topics of the recent NeurIPS conference. Since we can apply the same GNN graphs with different sizes, it is natural to ask how a GNN changes its behaviors for different graphs. Therefore, this research question takes the interest of the NeurIPS community.

Weaknesses: Soundness of the claims - In the introduction section (also in l.52, 186), the authors claim that we can use graph topologies as parameters. I wonder if the authors discussed the specific benefit of this interpretation. - Theory - We need to handle possibly unbounded operators on infinite-dimensional spaces when we deal with graphons. This paper reduces the problem to a finite dimension one by assuming the operator the Hilbert-Schmidt operator (hence the only accumulated point is 0) and that the filter function h is constant for $|h|<c$. The assumption on the filter function h means that a GNN passes the high-frequency component of signals. I think this is in contrast to the recent interpretation of a GNN as a low-pass filter (e.g., [NT and Maehara, 2019], [Oono and Suzuki, 2020]). I want the authors to discuss the feasibility of the assumption on the filter function. - [NT and Maehara, 2019] NT, H., & Maehara, T. (2019). Revisiting graph neural networks: All we have is low-pass filters. *arXiv preprint arXiv:1905.09550*. - [Oono and Suzuki, 2020] Oono, K & Suzuki, T (2020). Graph Neural Networks Exponentially Lose Expressive Power for Node Classification, ICLR 2020. - Experiment - It would be desirable if we can verify the trade-ff between the transferability and discriminability experimentally.

Correctness: Theory part: I did not find any critical mistake in describing the statements and their proofs. - The proof strategy seems reasonable. The authors evaluate two discretization errors. The first one is the discretization error of GNN operators. The other one is the discretization error of signals when discretizing Graphon into a graph with finite nodes. - l.230: the authors defined the convergent point of $delta_c$ by using $\mathcal{C}$. But I think it is not appropriate because $\mathcal{C}$ implicitly depends on $n$. When the multiplicity of some $\lambda_i$ is 1, the convergent point of $\delta_c$ is zero. In this case, it could worsen the dependence of the upper bound of Theorem 1 on $n$. Even worse, the bound could be vacuous. Can we guarantee the positivity of the convergent point (or what is the sufficient condition for the positivity)? Experiment part: I have several questions about whether the experiment results support the main claim. As stated at the beginning of Section 6, the purpose of the experiment is to explain Theorem2. So I evaluate the experiment from the viewpoint of consistency of it with Theorem2.) - The task is to assign the rating of the user 405. However, when creating the graph of users, they used the information of user 405 to connect similar users to the user 405. Since they used this connectivity information to predict the rating, I wonder if there is a possibility of information leakage. - I feel it is a bit strange to use RMSE since the task in the experiment is classification. I think it might be more natural for classification tasks to measure the difference in terms of the distance for probability distributions such as the KL divergence. Or it might be better to use a regression task, where RMSE is a natural distance.

Clarity: The authors properly prepared the necessary background information. The proof was not particularly hard to read.

Relation to Prior Work: Like this paper, the recent paper [Keriven et al., 20] derived the transferability of GNNs for graphs sampled from the same graphons. I want the authors to discuss the relationship between this paper and [Keriven et al., 20]. Differently from this paper, [Keriven et al., 20] stochastically samples graphs from a graphon. Their rate of the difference is $O((n\alpha_n)^{-1/2})$, where $\alpha_n$ is the parameter that determines the sparsity of graphs. When $\alpha_n$ is $O(1)$, the rate matches to this paper. [Keriven et al., 20] https://arxiv.org/abs/2006.01868

Reproducibility: Yes

Additional Feedback: 【Update after Authors Feedback】 I thank the authors for taking my comment into account seriously. The authors' feedback solved most of my questions. I agree with R3 in that the usage of graphon is somewhat unusual compared to other literature, in which we draw random graphs from a graphon. Therefore, I also recommend the authors to discuss the applicability of the theory. Considering the above evaluation into account, I keep my score (6). Feasibility of assumptions - First, my question confused the graph Laplacian and the adjacency matrix, as pointed by the authors. I am sorry for my misunderstanding. Still, I have a question. Let $W$ be the normalized adjacency matrix and $L=1-W$ is the normalized Lapalacian. Let $\lambda$, $\mu$ be their eigenvalues. Then, we ahve $0 \leq \mu\leq 2$ in general. Since $\lambda \approx 0$ means $\mu \approx 1$, it seems to me that $h = const$ on $|\lambda| < c$ corresponds to a filter that cuts components of a specific frequency range. - In L.156, the eigenvalue $\lambda_i$ of $T_W$ is claimed to between -1 and 1. However, I think this is not true since $W$ is not necessary the normalized adjacency matrix, although it is a typical example. If I do not miss anything, the proof does not use the assumption that $|\lambda_i| \leq 1$. Transferability–discriminability trade-off - I thank the authors' for explaining the trade-off. The authors' explanation is close to what I expected. Positivity of $\delta_c$ - I understand that we can ensure positivity of $\delta_c$ by correctly fixing the statement and proof. Related work on transferability - The authors pointed out that the result of [Keriven et al., 20] "depends on the graph sparsity". However, it seems the opposite to me because they assume dense graphs in the sense that the sparse parameter $\alpha_n$ should be $\Omega(\log n /n)$. - This paper is close to [Keriven et al., 20]. However, considering that this paper and [Keriven et al., 20] appear at almost the same time. The existence of [Keriven et al., 20] does not reduce the novelty of this paper. 【Initial Comment】 - l.107, 214: $\lambda_i^k$ is used as the $k$-th power of $\lambda_i$ in l.107, while it is used as the spectrum of $W_n$. I would suggest to avoid the same characters. - l.161: Mathematically, we do not have to treat $T_w^{(1)}$ as a special case (of course, it is OK to treat it as a special case as long as it is consitent with the inductive definition). - l.168: I wonder why the authors used WNN as the acronym of graphon neural network. - l.230: i should be made italic. - It is a kind of nitpicking. I think it is common to use a semicolon to separate input variables and function parameters in the argument list of a function. However, I have not seen a notation that uses two or more semicolons. So, I think $\Phi(\mathcal{H}; W, X)$ or $\Phi(X, W; \mathcal{H})$ is better if $W$ is considered as an input variable (or $\Phi(\mathcal{H}, W; X)$ or $\Phi(X; \mathcal{H}, W)$ if $W$ is a learnable parameter of a GNN.


Review 2

Summary and Contributions: The paper formalizes transferability of graph neural networks (GNN) based on the mathematical notion graphon. The analysis is designed for GNN acted on large graphs, due to the limiting nature of graphon. To my knowledge, it is the first work characterizing transferability of GNN using the graphon approach.

Strengths: The main theorem characterizing the limit behavior of GNN on large graphs is sound and novel. The follow-up numerical experiments on recommendation system shows the effectiveness of their theoretical bound.

Weaknesses: Experimental results are limited (only one), and not so typical for GNN applications. I believe people are more interested in empirical results on node classification, link prediction etc.

Correctness: Yes.

Clarity: Yes.

Relation to Prior Work: To my knowledge, it is the first work characterizing transferability of GNN using the graphon approach.

Reproducibility: Yes

Additional Feedback: The paper assumes that the graphs are constructed from graphon in a *fixed* manner, and this greatly limits the practical implications of its theoretical claim. Moreover, numerical experiments are not sufficient enough to complement the weakness of theoretical results.


Review 3

Summary and Contributions: This paper considers the behavior of graph neural networks on certain families of graphs (called graphons by the author, though the term is used in a non-standard way). The main result is that graph neural nets trained on certain deterministic sequences of graphs will converge to a certain limit object, with the particular implication that a graph neural network trained on one such sequence will perform well on a different instance of the sequence. ------------------ I've updated my score from 3->4 since there is enthusiasm among my co-reviewers. However, I think it is absolutely necessary for the paper to clarify immediately that the results don't apply to random graphs drawn from a graphon, but instead to deterministic weighted graph sequences derived from the graphon. Ideally, this should be accompanied by some discussion of why this setup should be informative about the behaviour of graph neural nets on graph data.

Strengths: The paper is very clearly written, the math seems tight (though I did not check it in detail), and the highest level question---when can a graph neural net trained in domain A transfer to domain B---is interesting.

Weaknesses: The major issue is that the version of the `graphon' model used in this paper doesn't seem relevant to practice, and seems unlikely to yield foundational insights about learning on graphs. The strategy is to generate graph sequences from graphons deterministically by returning completely connected weighted graphs with edge weights given by graphon values. The fact that a function (the graph neural net) on such sequences converges is not surprising, and doesn't seem to yield any deep insights.

Correctness: I believe the paper is technically correct. However, since my major issue with the paper is the lack of relevance of the theoretical model, I did not check the technical details.

Clarity: The paper is very clearly written.

Relation to Prior Work: The paper mentions several related papers on graph stability, though the connection is not explored in depth (this is understandable, given the page limit). The paper also cites a somewhat random subset of the graphon literature, though it's not clear to me how this literature relates to the results presented. The authors may be interested in http://proceedings.mlr.press/v89/veitch19a.html, which uses graphon methods (in the conventional sense) to study the behavior of graph embedding methods.

Reproducibility: Yes

Additional Feedback:


Review 4

Summary and Contributions: The paper introduces the notion of a graphon neural network, the limiting object os a graph neural network for large networks. THe authors leverage the theory of graphons, limiting objects of (dense) networks to introduce the idea of a graphon neural network, which serves the same role for GNNs. Their main result is a transferability result (Thm 2), showing that if two GNNs are associated to the same Graphon NN, the behave similar (their distance is bounded).

Strengths: The strongest contribution here is conceptual: by introducing the notion of a graphon NN the author enable comparisons across graph neural networks of the same class with a sound mathematical framework.

Weaknesses: * The weaknesses of the graphon framework are not discussed at all * The assumptions made in the proof in terms of graphon and signal properties could have been discussed more carefully * The experimental validation is weak

Correctness: The submission appears to be technically sound. The claims made by the authors are generally supported by analytical results. However, the authors are somewhat less careful when discussing the relative weaknesses and potential shortcomings of their approach. Specifically, it is well known that graphons are effectively limiting objects of *dense* graphs, which potentially limits there applicability in practice, as many networks of interest are (very) sparse. The authors should have at least discussed this point and potential strategies to deal with this problem. I see this as a clear shortcoming of the article. The computational validation of their results is also not very convincing: it is essentially a single experiment with real-world data that is performed here (there is a second set of experiments in the appendix but it is based on a quite simplistic setup: an SBM with two groups with a consensus dynamics (distributed averaging) on relatively small networks. I would have expected some more comparisons, with different datasets, different network structures etc. Moreover, why did the authors decide to scale the error bars by a factor 0.5 --- while stated in the caption, this is very misleading to a reader who may not carefully scan the caption and should be avoided in my opinion. Another point is that the various assumptions going into their proofs could have been discussed in a better manner, by clearly highlighting which limitations cannot really be circumvented, which are merely technical assumptions and what constrains cannot really be obtained.

Clarity: The paper is clearly written and organized, though I feel the authors could have moved some material to an appendix, and used the space in the main paper to expand on their results and discussion. For instance, some of the discussion on graphons as limiting objects is nice to know, but never used later on or required to understand the argument.

Relation to Prior Work: Related work is discussed appropriately.

Reproducibility: Yes

Additional Feedback: Typos and minor mistakes: - the degree matrix should be D = diag(A1) and not D=A1 (which would be a vector); moreover it is not a valid graph shift operator. - p.6 line 222: Lispchitz -> Lipschitz Following the discussion and authors' response I still see the paper as a good submission; even though the graphon model used is not a standard one and should probably be explained more clearly.

[Author Response · NeurIPS 2020]

We thank all reviewers for their time, effort and constructive feedback on this paper.

Experimental validation **(R1,R2,R4).** We appreciate the reviewers' concerns w.r.t. the experiments. We believe that
the contributions of this paper are primarily theoretical, namely, (i) showing that GNNs are transferable between
deterministic graphs obtained from a graphon and (ii) non-asymptotically quantifying this transfer error. Due to
space constraints, we only included limited numerical experiments to briefly illustrate these results in a more practical
context. Still, to address the reviewers' points, we will use the extra space in the camera-ready to include an additional
experiment using the citation network setting of (Kipf and Welling, 2017). We will also clarify in the manuscript that
the user network in the MovieLens example is built from training data alone as in (Ruiz et al., 2019b). The RMSE was
chosen for being a standard performance metric in collaborative filtering.

Feasibility of assumptions **(R1,R4).** To address the concerns of reviewer 4, we will add a discussion on the assumptions
of Thms. 1–2 to the camera-ready distinguishing between technical (e.g., normalized filters) and binding assumptions
(e.g., Lipschitz graphons) and highlighting their consequences and limitations. As for the point raised by reviewer 1,
note that the assumption on the filters $h$ is not contradicted by (NT and Maehara, 2019) and (Oono and Suzuki, 2020),
since they consider the spectrum of the graph Laplacian rather than the adjacency matrix. Hence, $|\lambda| \approx 1$ corresponds
to low frequencies and $|\lambda| \approx 0$ to high frequencies rather than the converse. Moreover, our results show that when the
relevant features have low frequency components ($|\lambda| \approx 1$), a simple graph convolutional filter is transferable (Thm. 3
in the appendices). This is not the case for high frequency features ($|\lambda| \approx 0$). In contrast, a GNN is likely to remain
transferable as it leverages nonlinearities to scatter high frequency components to lower frequencies.

Relevance and weaknesses of graphon framework **(R3, R4).** Reviewer 3 raises a good point in noting that the determin-
istic generative model we consider–which appears and is used throughout the seminal book on graphons (Lovász, 2012,
Chapter 10)–cannot produce certain types of sparse graphs (e.g., with fixed degree). However, it doesn't necessarily
yield complete graphs since $\mathbf{W}(u,v)$ can be 0 for some $(u,v)$. While we agree with the reviewer the model has
limitations, we believe that it still provides important insights into when and how transferability is possible. While it
may be intuitive that GNNs converge on sequences of deterministic graphs, this is not necessarily the case. For instance,
Thm. 3 shows that graph filters are only really transferable for large eigenvalues, i.e., "low frequency" features. This
is evidenced by the hypothesis that $h(\lambda)$ is constant for $|\lambda| < c$, which addresses the fact that the graph eigenvalues
accumulate near 0 and thus become harder to distinguish. GNNs, on the other hand, are expected to remain transferable
even for high frequency features ($|\lambda| \approx 0$) because the nonlinearities scatter high frequency components to lower
frequencies. These conclusions are not trivially obtained from the model. Naturally, as noted by reviewer 4, our analyses
focus on dense graphs for which graphons are models. Convergence of sparse graphs is a topic of active research that
requires considerably different entities (*graphings*) which are beyond the scope of this paper. Still, dense graphs have
practical applications and provide insights as to the transferability of the convolution operators in GNNs. We will make
these discussions more clear in the camera-ready version.

Transferability–discriminability trade-off **(R1).** This trade-off arises from $n_c$ and $\delta_c$, which are both related to the
threshold $c$. In the interval $|\lambda| \in [c, 1]$, the filters $h(\lambda)$ do not have to be constant and can thus distinguish between
eigenvalues. Hence, their discriminative power is larger for smaller $c$. The value of $n_c$ counts the eigenvalues $|\lambda| \in [c, 1]$.
When $c$ is small, $n_c$ is large, which worsens transfer error. The quantity $\delta_c$ is close to the minimum graphon eigengap
$min_{c \in [\lambda_i, \lambda_j]}|\lambda_i - \lambda_j|$. Since the graphon eigenvalues accumulate near 0, $\delta_c$ approaches 0 for small $c$. This also
increases the transfer error. Therefore, small values of $c$ provide better discriminability but deteriorate transferability,
and vice-versa. Note that for GNNs (Thm. 2) this trade-off is less rigid than for graph filters (Thm. 3 in the appendices)
because GNNs leverage nonlinearities to scatter low frequency components ($|\lambda| \approx 0$) to larger frequencies ($|\lambda| > c$).
These points will be added to the discussion of Thm. 2 in the paper. Finally, note that this trade-off is hard to verify
experimentally, since in practice $c$ can't be controlled as the filters $h$ are learned by training the GNN. How to promote
more high-pass filters during learning is an interesting avenue of research, but is beyond the scope of this paper.

Positivity of $\delta_c$ **(R1).** The reviewer raises an important point. The value of $\delta_c$ measures the minimum distance between
the *eigenspaces* corresponding to the closest graph eigenvalue $\lambda_i^n$ and graphon eigenvalue $\lambda_j$ on either side of $c$, i.e.
$\delta_c = min_{c \in [\lambda_i^n, \lambda_j]}|\lambda_i^n - \lambda_j|$ [cf. Prop. 2 in the appendices]. Since, $\lambda_i^n \to \lambda_i$, the convergent point of this quantity is
the difference between two consecutive distinct graphon eigenvalues, which is always positive. As defined in the paper,
however, $\delta_c$ only reflects this if $\lambda_i \neq \lambda_{i+\text{sgn}(i)}$ for all $i$, i.e., if all nonzero eigenvalues have multiplicity 1 and thus each
correspond to a distinct eigenspace. We will correct this definition to eliminate this requirement and clarify that $\delta_c$
converges to the smallest graphon eigengap $|\lambda_i - \lambda_j|$ such that $c \in [\lambda_i, \lambda_j]$, which is always positive for $c > 0$.

Related work on transferability **(R1).** We thank the reviewer for pointing us to (Keriven et al., 2020). In this paper, the
convergence of GNNs to continuous GNNs is studied based on concentration inequalities relating the spectrum of a
random graph Laplacian with that of the sampled graphs. Hence, their result depends on the graph sparsity. In contrast,
we focus on dense graphs associated with graphons and study GNN transferability by analyzing the spectral behavior of
graph convolutions. We will use the extra space in the camera-ready to expand on these distinctions.

[Meta-Review · NeurIPS 2020]

This article provides a theoretical analysis of the behaviour of graph neural networks (GNN), when applied to a specific deterministic sequence of graphs converging to a graphon. It gives bounds between the GNN outputs of a finite graph versus its graphon limit, and of the GNN output of graphs of different sizes. As such, it provides useful insights on GNNs, and about the transferability of parameters learned on a small graph to a larger graphs. An important point raised by R3, not really adressed in the response, needs to be adressed in the final version. The analysis is based on a deterministic sequence of graphs, and not on random graphs drawn from a graphon. Hence, the claim that a GNN trained on a single graph can be transferred to other graphs drawn from the same graphon using this result is incorrect. The authors should remove the following sentences, and clarify the contributions of the paper. Lines 4-5: "As a byproduct, coefficient can also be transferred to different graphs, thereby motivating the analysis of transferability across graphs." Lines 26-29: "In GNNs, there are two typical scenarios where transferability is desirable. The first involves applications in which we would like to reproduce a model trained on a graph to multiple other graphs without retraining. This would be the case, for instance, of replicating a GNN model for analysis of NYC air pollution data on the air pollution sensor network in Philadelphia." Lines 248-251: "This result is useful in two important ways. First, it means that, provided that its design parameters are chosen carefully, a GNN trained on a given graph can be transferred to multiple other graphs in the same graphon family with performance guarantees. This is desirable in problems where the same task has to be replicated on different networks, because it eliminates the need for retraining the GNN on every graph."